genomics/evolution/ecology

adaptation, population genetics, birds, colonization history, genome scan, spatial scales

**Authors for correspondence:**
Claudia A. Martin
e-mail: claudia.martin@uea.ac.uk
Lewis G. Spurgin
e-mail: l.spurgin@uea.ac.uk

# Genomic variation, population history and within-archipelago adaptation between island bird populations

Claudia A. Martin[1], Claire Armstrong[1,2], Juan Carlos Illera[3], Brent C. Emerson[4], David S. Richardson[1] and Lewis G. Spurgin[1]

[1]School of Biological Sciences, University of East Anglia, Norwich Research Park, Norwich NR4 7TJ, UK
[2]NERC Biomolecular Analysis Facility, Department of Animal and Plant Sciences, University of Sheffield, Alfred Denny Building, Western Bank, Sheffield S10 2TN, UK
[3]Oviedo University, Campus of Mieres, Research Unit of Biodiversity (UO-CSIC-PA), Research Building, 5th floor, c/Gonzalo Gutiérrez Quirós, s/n, 33600 Mieres, Asturias, Spain
[4]Island Ecology and Evolution Research Group, Institute of Natural Products and Agrobiology (IPNA-CSIC), C/Astrofísico Francisco Sánchez 3, 38206 La Laguna, Tenerife, Canary Islands, Spain

CAM, 0000-0003-2645-0790; CA, 0000-0003-3874-4243; JCI, 0000-0002-4389-0264; BCE, 0000-0003-4067-9858; DSR, 0000-0001-7226-9074; LGS, 0000-0002-0874-9281

Oceanic island archipelagos provide excellent models to understand evolutionary processes. Colonization events and gene flow can interact with selection to shape genetic variation at different spatial scales. Landscape-scale variation in biotic and abiotic factors may drive fine-scale selection within islands, while long-term evolutionary processes may drive divergence between distantly related populations. Here, we examine patterns of population history and selection between recently diverged populations of the Berthelot's pipit (*Anthus berthelotii*), a passerine endemic to three North Atlantic archipelagos. First, we use demographic trees and $f_3$ statistics to show that genome-wide divergence across the species range is largely shaped by colonization and bottlenecks, with evidence of very weak gene flow between populations. Then, using a genome scan approach, we identify signatures of divergent selection within archipelagos at single nucleotide polymorphisms (SNPs) in genes potentially associated with craniofacial development and DNA repair. We did not detect within-archipelago selection

at the same SNPs as were detected previously at broader spatial scales between archipelagos, but did identify signatures of selection at loci associated with similar biological functions. These findings suggest that similar ecological factors may repeatedly drive selection between recently separated populations, as well as at broad spatial scales across varied landscapes.

# 1. Introduction

Characterizing evolution at the genetic level is fundamental to our understanding of how populations adapt in response to changing ecological pressures [1]. The ability of species to adapt depends upon the amount of genetic diversity within populations, which in turn depends upon mutational processes, past and present demography, and selection. For a comprehensive understanding of how natural selection shapes genetic variation, studies are required on a variety of species with differing (and known) demographic histories, and across populations which have faced a wide range of selection pressures [2,3]. Studies on humans and laboratory model species have been important for our understanding of natural selection (e.g. [4,5]), but large-scale studies can now be carried out in most non-model organisms, providing opportunities for novel insights into evolutionary dynamics in the wild (see [6–8]).

Island archipelagos provide replicated, ecologically variable and simplified landscapes, that can greatly facilitate the study of adaptation in the wild ([9,10]). Varying abiotic environments, combined with independent evolutionary histories of inhabiting organisms, result in islands harbouring unique ecological communities [11–13]. The distinct geographical and ecological structure of individual islands, combined with the barrier to gene flow provided by the ocean, enables hierarchical population structure to develop over time, and for local adaptation to occur [14]. When combined with the large-scale genomic marker sets that can now be generated (e.g. [8,15]), island systems provide an excellent opportunity to tease apart the roles of selection, drift and gene flow in shaping patterns of genetic diversity in nature [16,17].

Selection operates at a range of geographical scales within and across island archipelagos [9]. Studying very fine landscape-scale adaptation within island populations may reveal ecologically relevant and rapid adaptation, but may be limited to detecting very strong signatures of selection [18–21]. Consequently, studies of fine-scale local adaptation may be biased towards detecting phenotypes determined by genes of large effect, while smaller effect loci or highly polygenic phenotypes are likely to go undetected [22]. Furthermore, local adaptation may be transient, as a temporary response to fluctuating selection pressures, and therefore, patterns of adaptation at one timepoint may not be relevant to longer-term evolutionary processes. By contrast, selection can also be studied at broad spatial scales, among island archipelagos between which there has been long-term isolation and limited gene flow [23,24]. Such studies may reveal patterns of strong adaptive evolution [25], but signatures of selection may be eroded by subsequent evolutionary forces including mutation, drift and gene flow that accumulate over time [26,27]. It has long been recognized that consideration of spatial scales is important when identifying patterns and drivers of adaptation among populations, but few studies have quantified adaptation across a range of scales. In particular, it is important not to neglect intermediate scales (e.g. between populations on closely located islands with recent divergence histories and/or potential for gene flow) when studying adaptation, as these may provide powerful systems with which to detect ecologically relevant adaptations.

Berthelot's pipit (*Anthus berthelotii*) is a Macaronesian endemic passerine distributed across three North Atlantic archipelagos (figure 1). Previous research suggests that this species initially colonized the Canary Islands from mainland Africa approximately 2.5 Ma [28], before dispersing independently from the Canary Islands to both the Selvagens and the Madeiran archipelago, approximately 8500 years ago [29]. These founder events resulted in population bottlenecks and reduced population size across the northward colonized archipelagos, with a subsequent absence of gene flow between archipelagos [29–31]. Founder effects across archipelagos appear to shape genetic and morphological divergence of populations at broad scales [29,31]. Little is known about divergence, or levels of migration, between populations within archipelagos. Selective pressures including diseases and climatic factors vary greatly across these populations, both at broad geographical scales between archipelagos and at finer geographical scales between and within islands [32]. For example, pathogen prevalence (i.e. *Avipoxvirus* and *Plasmodium*) varies greatly among islands within archipelagos. Both the Canary Islands and Madeiran archipelago have populations with both high and low pathogen loads, and population-level patterns of pathogen prevalence are consistent over time [33]. Broad-scale balancing selection appears to have maintained variation at an important immune gene

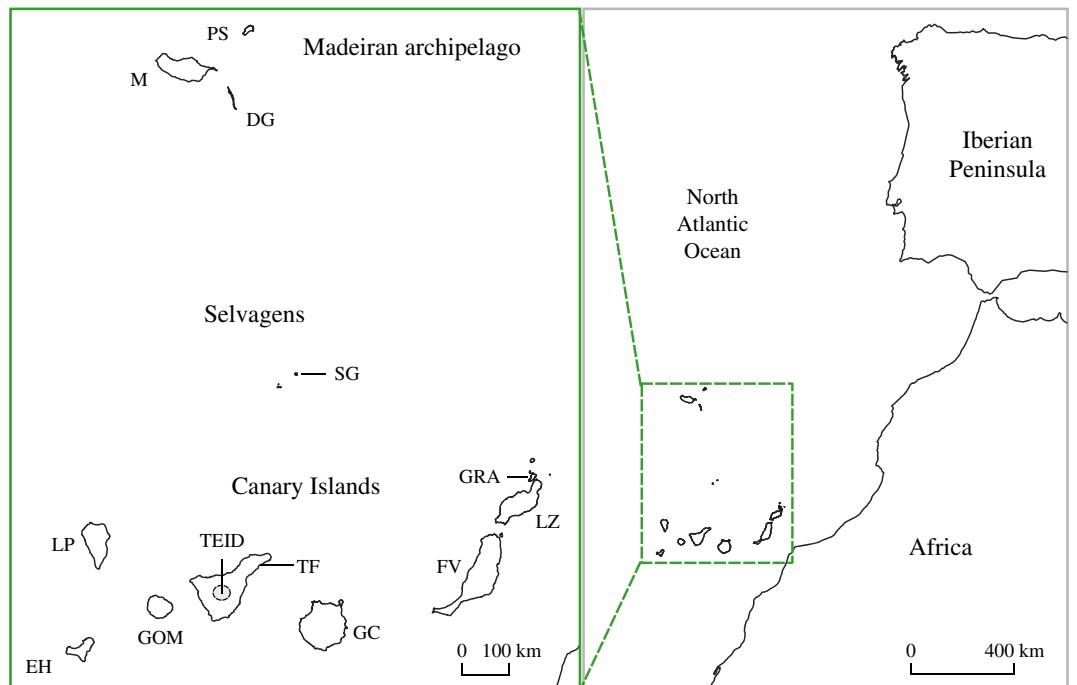

**Figure 1.** Locations of Berthelot's pipit populations used in the current study. Canary Island populations: El Hierro (EH), La Palma (LP), La Gomera (GOM), El Teide (TEID) mountain population of Tenerife, lowland Tenerife (TF), Gran Canaria (GC), Fuerteventura (FV), Lanzarote (LZ) and La Graciosa (GRA). Madeiran populations: Madeira (M), Porto Santo (PS) and Deserta Grande (DG). Selvagem Grande (SG), Selvagens archipelago.

family, the major histocompatibility complex across archipelagos [34]. Selection also operates over very fine spatial scales within this system, with previous work having identified landscape-level environmental drivers of pathogen distribution and immunogenetic variation within specific islands [35–37]. Climatic conditions, and rainfall in particular, also vary strongly between western and eastern Canary Islands, and between Madeiran islands [38]. Thus, the Berthelot's pipit system provides an excellent framework with which to investigate how natural selection operates across different spatial scales in nature.

A recent study by Armstrong *et al.* [31] used a genome-wide set of markers to investigate genetic variation and selection at broad spatial scales across the Berthelot's pipit system—specifically between the three archipelagos. Analysis showed strong genetic structure among, but not within, archipelagos, while a genome scan to identify loci under selection between archipelagos identified candidate genes associated with bill morphology, immunity and adaptation to climate. However, we do not yet understand (i) patterns of colonization, gene flow and drift within archipelagos, and (ii) whether the same loci and/or traits showing divergent selection between archipelagos are also under selection between recently separated populations within archipelagos. Such information will provide useful insight into how selection operates across different spatial scales in this and other systems.

Here, we use genomic approaches to investigate population history and genetic diversity across island populations of Berthelot's pipit, and test for signatures of selection between recently separated populations within archipelagos. We use genome-wide restriction-site associated DNA sequenced (RAD-seq) markers from across the Berthelot's pipit range to address the following questions: (i) what new insights do analyses of genomic variation provide for population history, including colonization, bottlenecks and gene flow across the species range? (ii) can we detect signatures of selection across recently diverged populations within archipelagos? (iii) what are the loci, and traits, under selection within archipelagos? and (iv) are the same loci under selection within and across archipelagos? To address these questions, we first use population genetic analyses to examine colonization history and gene flow across the species range. We also quantify population structure and genetic diversity within the Canarian and Madeiran archipelagos independently, providing a finer-scale assessment of genetic structure compared to previous studies in this system. We then use genome scan approaches to identify loci under divergent selection within archipelagos, and where appropriate, link patterns of genetic variation to variation in phenotypic traits. Finally, we compare our results to previous research

on this species, to help understand how population history, selection and drift interact to shape patterns of diversity at different scales across island populations.

# 2. Methods

## 2.1. Population sampling and sequencing

Berthelot's pipits were sampled on 12 islands across their geographical range (figure 1), as reported in detail by Illera *et al.* [30] and Spurgin *et al.* [33]. As in Armstrong *et al.* [31], we consider the pipits inhabiting El Teide mountain plateau of Tenerife (greater than 2000 m above sea level) as a separate population to that inhabiting the island's lowlands owing to their separation by a wide strip of forest vegetation on the mountain side which the pipits do not inhabit. Individuals were sampled widely across the populations, reducing the probability of sampling closely related individuals, and caught using spring traps baited with *Tenebrio molitor* larvae. A blood sample (*ca* 25 µl) was taken from each bird by brachial venipuncture and stored in 800 µl absolute ethanol at room temperature. DNA was extracted using the salt extraction protocol described by Richardson *et al.* [39], and birds were molecularly sexed [40]. Seven morphometric measurements were taken; weight, wing, head and tarsus length, and bill height, length and width. Each individual was fitted with a colour or metal ring to prevent resampling of the same individuals. Birds were released unharmed at the point they were captured. Twenty putatively unrelated individuals were selected from each population (22 from the lowland Tenerife population) for double digest RAD-seq (ddRAD-seq), with efforts made to equalize the sex ratio within each population sample [31].

The initial ddRAD library was generated using the protocol by DaCosta & Sorenson [41] which assigns RAD reads to samples based on an 8 bp barcode sequence and retains the read with the highest quality score. Loci that could not be confidently genotyped in more than four samples and those where 10% or more calls were missing or ambiguous were treated as missing data in the 'Berthelot's' library. The 'All Pipits' dataset containing all Berthelot's pipit and tawny pipit samples was filtered to contain single nucleotide polymorphisms (SNPs) from RAD loci that were successfully genotyped in 100% of individuals, removing loci that contained SNPs with greater than two alleles. RAD loci were mapped to the zebra finch genome (*Taeniopygia guttata*; v. 3.2.4; [42]). The data included multiple SNPs originating from the same RAD loci (throughout, marker names refer to distance in bp from the start of the RAD tag: 'Locus number–s–bp from start').

The Berthelot's marker sets were first grouped by archipelago and then trimmed using Plink 1.9 [43] to remove sex-linked loci and SNPs with low minor allele frequency (MAF < 0.03) with the aim of removing exceptionally rare variants within archipelagos while retaining a large marker set (MAF threshold reviewed by Linck & Battey [44]). We used Plink and GCTA v. 1.91.7 [45] to calculate genetic relatedness between each pair of individuals (dyad) for each of the populations. GCTA relatedness values were strongly correlated with those calculated by Plink (Pearson correlation; $r = 0.92$, 0.98 and 0.96 for the Canary Islands, Madeiran archipelago and Selvagens, respectively), so we only report the Plink calculated values. Using these (electronic supplementary material, figure S1), one individual from any pair identified as having a relatedness value greater than 0.2 was randomly removed to avoid first- and second-order relatives being included in the population genetic and selection analyses.

## 2.2. Inferring population divergence, admixture and genetic diversity

Population genetic analyses were carried out using two datasets to determine patterns of colonization and gene flow among populations across the species range (figure 1). Strong population structuring exists between archipelagos of the Berthelot's pipit, supporting our previous inference of the absence of contemporary gene flow at broad scales in this system [31]. We have reported weak east–west population structure between populations within the Canary Islands [31], but it is unknown whether this is ownig to contemporary gene flow between closely located islands or weak population divergence since colonization. We implemented *TreeMix* at these different population scales with the aim of further understanding the evolutionary processes behind the patterns of population structuring we see. First, we used an 'All Pipits' dataset which, in addition to the Berthelot's pipits includes 16 tawny pipits (*Anthus campestris*), the Berthelot's pipit sister species [28], sampled from northeast African and Spanish populations. The tawny pipit was used to root divergence from the mainland across the three Macaronesian archipelagos, to gain insight into the earliest colonized islands. Full details for this dataset and specific parameters used are in the electronic supplementary material,

Methods and table S1. Second, we used the 'Berthelot's' dataset, as described above, which includes only Berthelot's pipits. This dataset provides a greater number of polymorphic loci within the Berthelot's pipit populations owing to lower within-species divergence which may enhance the ability to detect population splits, migration and drift among populations. For this analysis, we trimmed the marker sets across all 13 populations using Plink, to remove closely related individuals (as above) and loci with MAF < 0.03, loci in strong linkage disequilibrium (LD) (greater than $0.4r^2$ threshold, for a sliding window 50 kb with 10 marker step) and sex-linked loci (electronic supplementary material, table S1).

Using TreeMix v. 1.13 [46], we inferred a tree in which populations (i.e. one population per island except in Tenerife with two populations, one in the lowlands and one in the highlands) may maintain gene flow after they split from a common ancestor. This method first infers a maximum-likelihood tree from genome-wide allele frequencies and then identifies populations with poor fit to this model (populations with residuals deviating strongly from zero); migration events involving these populations are added in order to improve the likelihood of the model. Allele frequencies for TreeMix analysis were calculated within populations using Plink, after marker pruning (electronic supplementary material, table S1). We modelled several scenarios allowing zero to eight migration events (electronic supplementary material, table S1), discounting migration events when the relative increase in model likelihood was less than 1%. For each analyses, 10 000 bootstrap replicates were generated, resampling blocks of 20, 50 and 80 SNPs to evaluate the robustness of the tree topology; this corresponds to a window size of approximately 10–30 Mb as used by Pickrell & Pritchard [46]. The total fraction of the variance explained by each model was estimated with the 'get_f()' R function, in TreeMix. Residual plots were assessed to display model fit and identify poorly fitted population pairs. $F_{ST}$ was calculated between pairs of populations in Plink [47] using the genome-wide RAD dataset as trimmed for the 'All Pipits' and 'Berthelot's' tree to support TreeMix tree topology. To test for admixture among Berthelot's pipit populations, we computed the three-population statistic ($f_3$ statistic; [48]) for all population triplets through software threepop [49] implemented in TreeMix, jackknifing over blocks of 50 SNPs. An observed negative value of the $f_3$ statistic and Z-score < −2 are indicative of historical admixture [49].

We next investigated fine-scale population genetic structure between recently separated populations within archipelagos. From the initial Berthelot's RAD library, we generated separate Canary Islands and Madeiran archipelago marker sets prior to trimming to maximize the number of loci at each level of clustering within-archipelago datasets (electronic supplementary material, table S1). As we only sampled one population in the Selvagens, no within-archipelago analysis was conducted for this archipelago. These data were also trimmed to remove SNPs with MAF < 0.03 (principal component analysis (PCA) was filtered according to MAF (SNPs with MAF < 0.03 excluded) within archipelagos, LD analysis was MAF-filtered (SNPs with MAF < 0.03 excluded) within populations), and closely related individuals were removed (as above). LD summarizes both mutational and recombination history, whereby larger, more outbred populations show rapid decay of LD between genetic markers compared to small inbred populations [50]. Patterns of LD have been used extensively to detect historic fluctuations in population size ($N_e$) and founder events in humans [49,51], selectively bred species such as Chinese Merino sheep, Xinjiang type [52] and wild species including European grey wolves, Canis lupus [25], and village dogs, Canis lupus familiaris [53]. The relationship between proximate SNPs reflects historic $N_e$, and LD at distant SNPs reflects $N_e$ in more recent time. To further understand patterns of genetic diversity and population size in the Berthelot's pipit, we estimated LD for each island population using Plink. The $r^2$ values were compared to physical distance between loci for all pairs of SNPs situated on the same chromosome. We fitted a locally weighted linear regression (loess) curve to the $r^2$ data using the R function 'loess' using the default span parameter (0.75), with 95% confidence intervals calculated. Population structure was examined within the Canary Islands and Madeiran archipelago independently using a PCA, implemented using Plink, based on the trimmed and filtered marker sets.

## 2.3. Genome scan for signatures of selection within archipelagos

For genome scan analyses, using the archipelago level marker sets, close relatives and SNPs with an MAF < 0.03 were removed (as above), but we did not filter based on LD, which enabled us to identify and visualize genomic regions under selection (electronic supplementary material, table S1). We used EIGENGWAS [54], implemented in the program GEAR (www.github.com/gc5k/GEAR/wiki), to identify loci consistent with selection within archipelagos. EIGENGWAS performs a PCA to generate gradients of population structure, then assesses each genetic marker individually for an association with these axes. EIGENGWAS provides genomic inflation factor corrected p-values ($\lambda_{GC}$)—with the significance threshold determined by

Bonferroni-correction—to control for genome-wide population stratification and drift. Loci above this significance threshold are putatively under selection across the gradient of population structure (see PCAs, figure 3). We also calculated $F_{ST}$ for each SNP, using all SNPs that had passed the trimming stages (electronic supplementary material, table S1). All SNPs in the Madeiran subset were also in the Canary Island dataset as a result of reduced genetic diversity in the Madeiran archipelago, and hence, direct comparisons of SNP variation are made. To identify genes located near outlier SNPs, we viewed regions of interest using the zebra finch genome (v. Taeniopygia_guttata-3.2.4) in NCBI Genome Data Viewer v. 4.8. (www.ncbi.nlm.nih.gov/genome/gdv/browser).

After having identified candidate SNPs which may be associated with skeletal development (see Results/Discussion), we determined the genotype–phenotype associations for these loci across all Berthelot's pipit populations. We used linear mixed models (LMMs) implemented in R using the lme4 package v. 1.1.15, to test the hypothesis that SNP variation within the candidate gene is associated with phenotypic variation in morphological traits. To check whether skeletal development is associated with genotypic variation at these candidate SNPs, we fitted LMMs with wing, tarsus and head length, bill length, width and height and weight as dependant variables. Population was modelled as a random effect nested within archipelago, and genotype (number of copies of the minor allele), sex and age as fixed effects in the model. Separate models were fitted for each of the SNPs within the morphology associated gene. All estimates are reported with associated 95% confidence intervals.

# 3. Results

The initial RAD library provided 9960 genome-wide polymorphic SNPs across the entire geographical population range of Berthelot's pipit. After MAF filtering separately in each archipelago, we retained 4470 SNPs in the Canary Islands subset and 2938 in the Madeiran archipelago subset. For the *TreeMix* Berthelot's tree, we trimmed all 13 Berthelot's pipit populations together from the initial RAD library, and after MAF and LD filtering retained 2850 loci across the Canary Islands, Madeiran archipelago and Selvagens.

Relatedness varied within and among populations: while most pairs of individuals showed low relatedness ($r < 0.05$), there were some pairs with high relatedness sampled in the smallest isolated populations of the Canary Islands (El Hierro and La Graciosa) and the islands of Madeira and Deserta Grande in the Madeiran archipelago (electronic supplementary material, figure S1). No pairs of individuals had relatedness of $r > 0.2$ in the Selvagens. To avoid including closely related individuals in the population genetic and selection analyses, one individual was removed from each dyad with a relatedness score $r > 0.2$, resulting in three individuals from La Graciosa, one from Lanzarote, two from El Hierro, three from Madeira and seven from Deserta Grande being removed.

## 3.1. Population genetic analyses

We used *TreeMix* to produce maximum-likelihood trees of divergence and gene flow. The tawny pipit rooted tree showed strong divergence between mainland tawny pipits and the contemporary Berthelot's pipit populations ($F_{ST}$; Canary Islands 0.36–0.38, Madeiran archipelago 0.42–0.44 and Selvagens 0.46), with shortest branch lengths to the central Canary Islands (electronic supplementary material, figure S2A). Populations in the Madeiran archipelago had poor residual fit with the tawny pipit (electronic supplementary material, figure S2), suggesting that these populations may be more closely related than is presented by the best-fit tree. However, adding migration events did not improve the residual fit of these models. Despite this, the tawny pipit rooted tree, obtained without adding migration events, explained the majority of allele frequency variation (variance = 99.55%) between populations.

Maximum-likelihood trees limited to the 13 Berthelot's pipit populations placed the Madeiran and Selvagens archipelagos on long, independent branches, grouping with the central/eastern and western Canary Islands, respectively (figure 2a). Tree topology within the Canary Islands roughly reflects geographical distance between islands, with branches east and west of Tenerife, and Mount Teide as a separate branch point from lowland Tenerife (figure 2a). Generally, weak drift is observed across the Canary Islands, with the longest within-archipelago branch lengths in El Hierro and La Graciosa. These same patterns are reflected in pairwise $F_{ST}$ values, with moderate divergence between El Hierro and La Graciosa ($F_{ST} = 0.04$) the most geographically distant pair of populations, and increasingly weaker divergence between more closely located populations in the Canary Islands, especially in the central islands in the archipelago ($F_{ST}$ range = 0.01–0.03). In the Madeiran archipelago, longer branch lengths

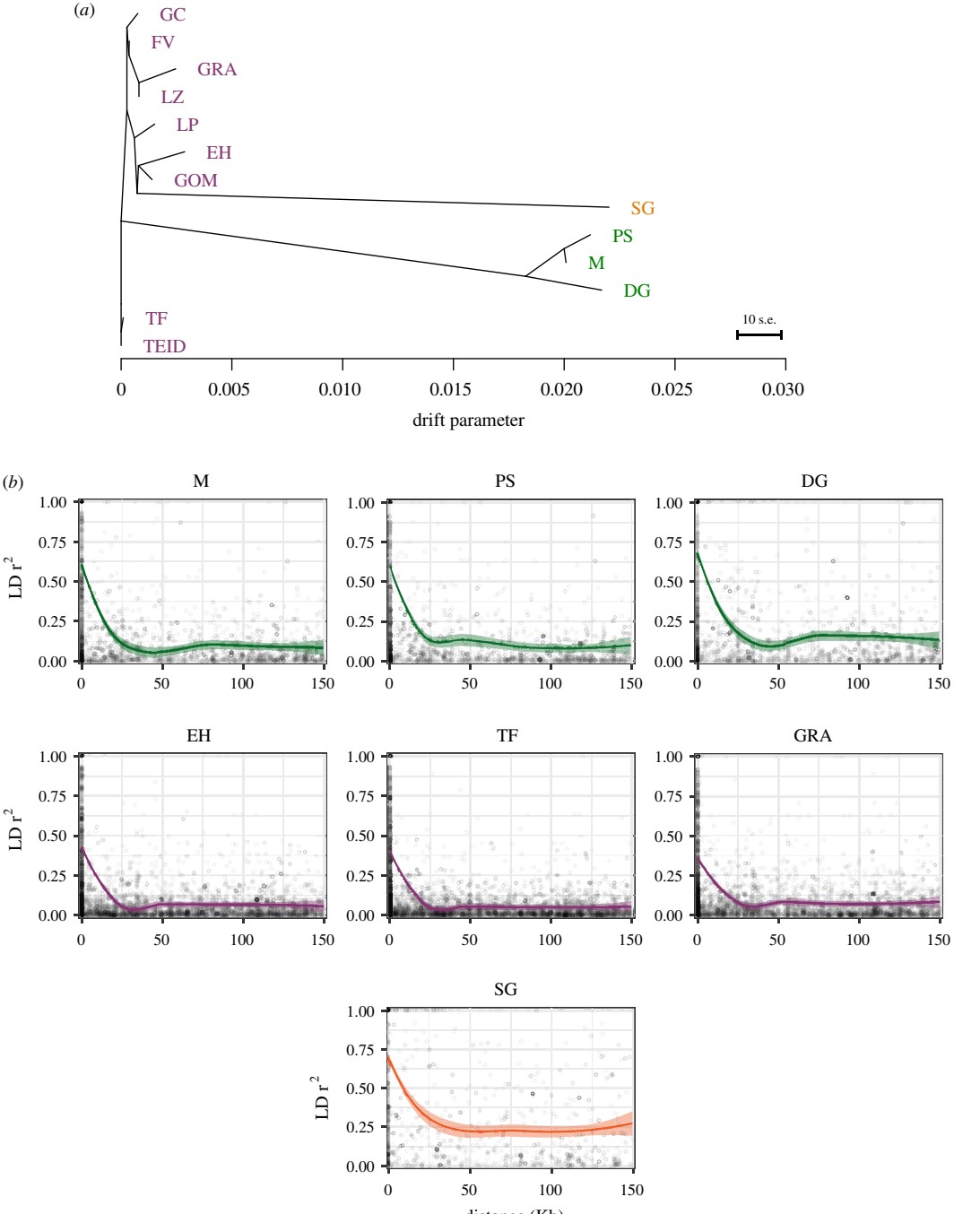

**Figure 2.** Evolutionary relationships between island populations of the Berthelot's pipit. (a) Maximum-likelihood bifurcating tree of population history—without subsequent gene flow—across the pipit colonization range as inferred by *TreeMix*. The branch length-scale bar shows 10 times the average standard error in the covariance matrix of ancestry. (b) The relationship between LD and base-pair distance for SNPs across each Madeiran Island (green), three Canary Island populations (purple), and the Selvagens (orange); Tenerife, a central island assumed to be a large outbred population with low within-archipelago divergence (figure 3a); and El Hierro and La Graciosa populations which have long branch lengths and strongest within-archipelago genome-wide divergence. The fit lines show a local regression model, with a shaded band indicating 95% confidence intervals.

were observed, with the highest divergence between Deserta Grande and the other Madeiran islands ($F_{ST}$; Porto Santo = 0.06, Madeira = 0.05). Tree topology was broadly robust to window size, but there were minor differences within the Canary Islands including the source populations for the Madeiran archipelago (electronic supplementary material, figure S3); here, we present specific model results calculated using windows of 50 SNPs. The majority of allele frequency variation (variance = 99.86%) is described solely by the tree topology, with good residual support for most populations (electronic

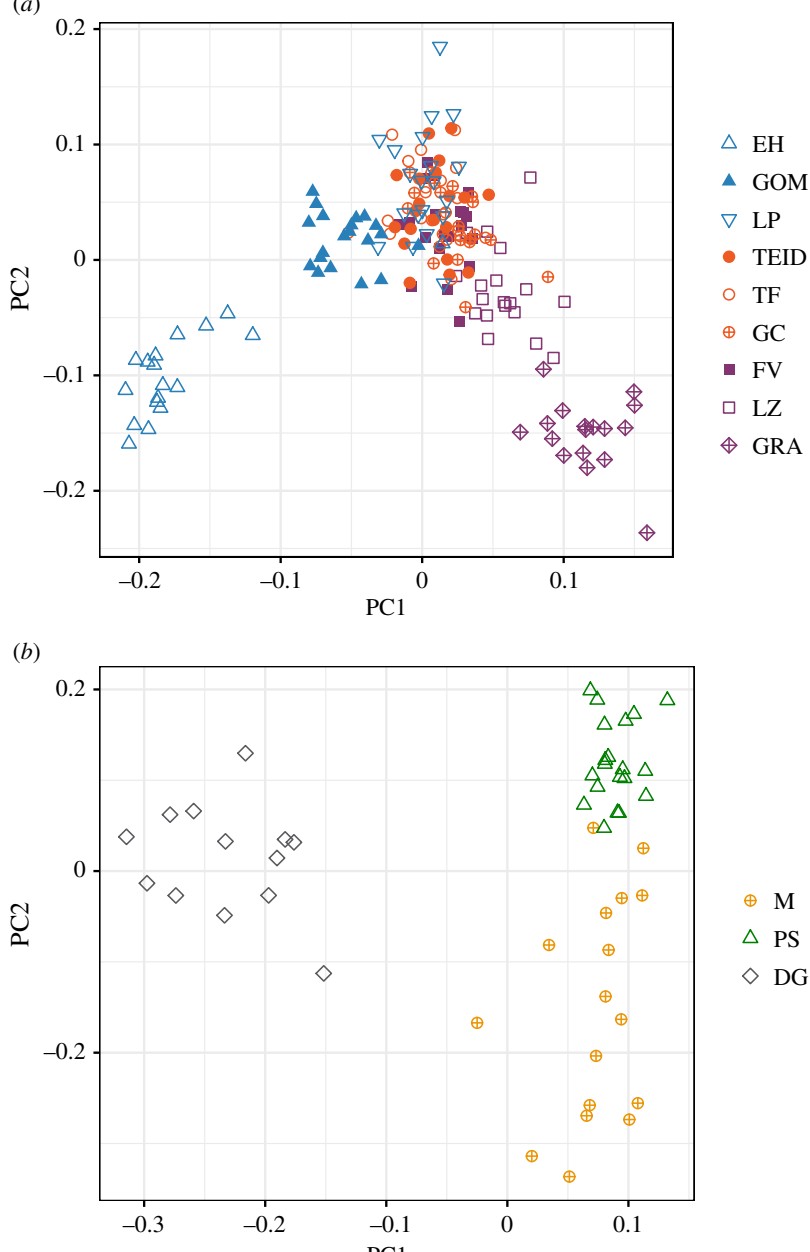

**Figure 3.** Population structure based on genome-wide ddRAD SNPs among Berthelot's pipit populations separately across the Canary Islands and Madeiran archipelago. (*a*) PCA across the Canary Island populations. PC1 and PC2 explained 2.7% and 2.3% of genomic variation, respectively. (*b*) PCA of Madeiran archipelago populations; PC1 = 3.5%, PC2 = 1.6% of genomic variation explained. Canary Island populations: El Hierro (EH), La Palma (LP), La Gomera (GOM), El Teide (TEID), lowland Tenerife (TF), Gran Canaria (GC), Fuerteventura (FV), Lanzarote (LZ) and La Graciosa (GRA). Madeiran populations: Madeira (M), Porto Santo (PS) and Deserta Grande (DG).

supplementary material, figure S4). Sequentially adding migration events did not substantially improve model support (electronic supplementary material, figure S5) or the degree of variance explained by trees (increase in variance = 0.11% after five migration events added; electronic supplementary material, figure S6). Weakly negative $f_3$ statistics (>− $5.3 \times 10^{-4}$) and Z-scores > −1.3 were found for Tenerife (including El Teide), Gran Canaria, Fuerteventura and Lanzarote in the Canary Islands and the island of Madeira when in a three-way population comparison (electronic supplementary material, table S2). $F_3$ results suggest few admixture events subsequent to branch divergence may have occurred between Madeira and Porto Santo and Fuerteventura/Lanzarote and La Graciosa. This is consistent with geographical distance between islands, suggesting no admixture between geographically distant populations.

Patterns of LD within populations are shown in figure 2*b* and electronic supplementary material, figure S7. LD was highest in the smallest and most isolated populations across the Berthelot's pipit range. Thus, across all populations, the Selvagens had the highest LD with a long-range decay pattern; LD was lower across the three Madeiran islands, and was lowest in the Canary Islands, especially in large central islands (figure 2*b*).

To investigate potential-fine-scale population structure within archipelagos, we conducted a PCA of individuals using the archipelago datasets. Within the Canary Islands, we found that the first principal component (PC) roughly reflects an east–west gradient of population structure, with El Hierro and La Graciosa separating most distinctly from the other islands (figure 3*a*). The second component separates El Hierro, Lanzarote and La Graciosa from the other islands. PCA of the Madeiran archipelago separated Deserta Grande from Madeira and Porto Santo along the first axis, and separated Porto Santo from Madeira on the second axis, with a weak gradient from Madeira to Deserta Grande to Porto Santo (figure 3*b*).

## 3.2. Genome scan to identify loci under selection within archipelagos

We used EIGENGWAS analyses to identify loci under divergent selection across the gradients of population structure seen in the Canary Islands and Madeiran archipelago, separately (figure 3). Across the Canary Islands, genomic inflation factors ($\lambda_{GC}$) were substantial for both PCs (PC1 = 2.4 and PC2 = 2.1, where a value greater than 1 indicates population structuring; [55]). Correcting for this and using a Bonferroni-corrected significance of $p < 1.12 \times 10^{-5}$ (*n* SNPs = 4470) for the Canary Islands, we detected one outlier SNP for PC1 ($p = 3.56 \times 10^{-9}$, figure 4*a*; electronic supplementary material, figure S8A). No outlier SNPs were detected for PC2 (electronic supplementary material, figure S9A,B). For the Madeiran archipelago, after correcting for the $\lambda_{GC}$ of 3.4 for PC1 and 1.4 for PC2, three outlier SNPs exceeded our Bonferroni-corrected significance threshold of $p < 1.70 \times 10^{-5}$ (*n* SNPs = 2938). Two of these were on the same RAD locus, within 20 bases of each other (figure 4*b*; electronic supplementary material, figure S8B; table 1). No outlier loci were detected for PC2 across Madeira (electronic supplementary material, figure S9C,D). Allele frequencies for all loci detected in the EIGENGWAS analyses are reported in the electronic supplementary material, table S3.

Locus $F_{ST}$ values were not correlated between archipelagos (Spearman's-rank correlation, $r = 0.031$, $p = 0.184$), but significant SNPs from the EIGENGWAS analyses had the highest $F_{ST}$ values (figure 5). The SNP detected by EIGENGWAS as being under selection across the Canary Islands (219s24), had a high MAF in the western islands of El Hierro and La Palma, and a low MAF across the central and eastern islands within that archipelago (electronic supplementary material, table S3). All SNPs under selection across the Madeiran archipelago had near 50% prevalence of the 'minor allele' in the Deserta Grande population while being absent from the two other islands, with a low frequency of the minor allele observed across the Canary Islands and Selvagens.

We were able to map all significant EIGENGWAS SNPs to the zebra finch genome and determine their likely genomic location (table 1; electronic supplementary material, figure S10). The two closest genes to the Canary Island SNP, 219s24, are *WDHD1* and *GCH1*, which are involved in DNA binding/repair and enzyme synthesis, respectively (table 1). This SNP maps to chromosome 5, with *WDHD1* 2071 bases upstream and *GCH1* 6252 bases downstream. In the Madeiran archipelago, the two significant outlier SNPs in the same RAD locus (1585s94 and 1585s112) mapped to intronic regions of a candidate gene for morphology, *ADAM12*, on chromosome 6 (see Discussion; table 1). The third SNP, 790s54, was not close to a gene (closest gene 70 799 bp downstream).

## 3.3. Genotype–phenotype association across populations

*ADAM12* has been shown to play a role in skeletal development and is, therefore, a potential candidate for being associated with morphology (table 1, see Discussion). Using LMMs, we tested how variation at this locus was related to candidate morphology traits across all pipit populations. To determine what effect candidate SNP variation may have on morphology, we tested for genotype associations with wing, tarsus and head length, weight and bill length, width and height. Genotypes for SNP 1585s94 and 1585s112 within the *ADAM12* gene were strongly colinear ($R^2 > 0.982$). We found a similar effect of genotype on head length at both SNPs putatively under selection within the *ADAM12* gene (Gaussian LMM, SNP 1585s112 estimate ± s.e. = −0.39 ± 0.13, $p = 0.003$; $R^2 = 0.75$; electronic supplementary material, figure S11 and SNP 1585s94 estimate ± s.e. = −0.36 ± 0.13, $p = 0.006$; $R^2 = 0.75$) as well as strong differences between the sexes ($p < 3.3 \times 10^{-6}$). Homozygous individuals for the minor allele were only

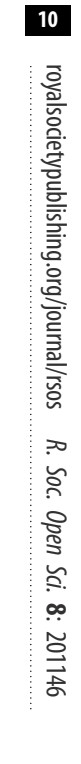

**Figure 4.** Manhattan plots of EigenGWAS analyses based on genome-wide ddRAD SNPs among Berthelot's pipit populations within archipelagos. (*a*) Canary Islands PC1, clustering east–west geographical gradient, as seen in figure 3*a*. Horizontal red line indicates Bonferroni-corrected significance of $p < 1.12 \times 10^{-5}$ based on 4470 genome-wide ddRAD SNPs. (*b*) Madeiran archipelago PC1, separating Deserta Grande from Madeira and Porto Santo islands, as seen in figure 3*b*. Horizontal red line indicates Bonferroni-corrected significance of $p < 1.70 \times 10^{-5}$ based on 2938 genome-wide ddRAD SNPs. Unmapped SNPs are recorded as 'Un', and alternate black-grey colouring indicates chromosomal limits.

detected in the Deserta Grande population for both of the SNPs, while heterozygous individuals were present at low frequency for eight the 13 pipit populations (electronic supplementary material, figure S11 and table S3). Genotype was not significantly associated with beak morphology variables (bill length, height or width), weight, wing length or tarsus length, although there were differences between sexes for tarsus length ($p < 0.003$), wing length ($p < 0.002$) and bill length ($p < 0.010$) as we expect for a sexually dimorphic species.

# 4. Discussion

We examined genetic divergence and selection between recently founded island populations in an attempt to understand population history and uncover traits of adaptive importance across selective environments in the wild. Using RAD sequenced markers generated for 13 populations of Berthelot's pipit, we first analysed genome-wide variation to uncover patterns of colonization, admixture and population demography. Our analyses support the establishment of Berthelot's pipit across the three archipelagos via independent colonization events, with evidence of weak subsequent gene flow between populations. Patterns of genetic diversity are consistent with signatures of founder events and geographical isolation. We applied a genome scan approach to identify signatures of selection,

**Table 1.** Outlier SNPs identified by EIGENGWAS analyses across the Canary Islands and/or Madeiran archipelago populations of Berthelot's pipit (figure 4). (Genes within 10 000 bp of the SNP are identified. Relative positions of the candidate genes are stated in bp upstream (US) or downstream (DS) from the SNP site.)

| SNP | $P$ Canary Islands/ $P$ Madeira | $F_{ST}$ Canary Islands/ $F_{ST}$ Madeira | genomic location | candidate gene(s) | gene product | trait |
|---|---|---|---|---|---|---|
| 219s24 | $3.56 \times 10^{-9}$/— | 0.23/0 | Chr 5: 58990367 | WDHD1 (2071 US) | WD repeat and HMG-box DNA binding protein 1 | DNA binding and repair [56]. |
| | | | | GCH1 (6252 DS) | GTP cyclohydrolase 1 | Rate-limiting enzyme for tetrahydrobiopter-in (BH4), a vital cofactor and modulator of peripheral neuropathic and inflammatory pain [57] |
| 158s594 | $0.61/1.59 \times 10^{-6}$ | 0.02/0.56 | Chr 6: 33504910 | ADAM12 (in gene) | disintegrin and metalloprotease domain 12 | body size by affecting bone/cartilage development [58] |
| 158s5112 | $0.65/1.59 \times 10^{-6}$ | 0.02/0.56 | Chr 6: 33504928 | ADAM12 (in gene) | disintegrin and metalloprotease domain 12 | body size by affecting bone/cartilage development [58] |
| 790s54 | $0.31/1.16 \times 10^{-5}$ | 0.01/0.47 | Chr 24: 718487 | — | — | — |

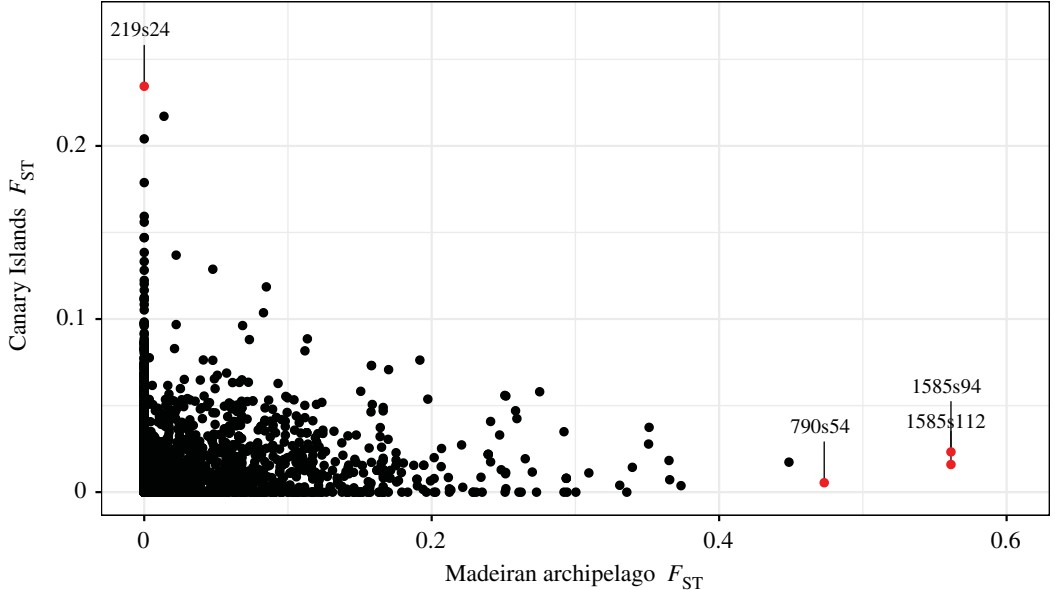

**Figure 5.** Within-archipelago genetic differentiation of genome-wide ddRAD SNPs among Berthelot's pipit populations. Points represent $F_{ST}$ of 3531 mapped genome-wide ddRAD SNPs between populations, across the Canary Islands and the Madeiran archipelago. SNPs identified by the EigenGWAS analysis for the Canary Islands archipelago (y-axis) and Madeiran archipelago (x-axis) are highlighted in red and labelled with their SNP code. Please note: no unmapped SNPs had $F_{ST} > 0.16$ or greater than 0.40 for the Canary Islands and Madeiran archipelago, respectively.

and inferred traits of ecological importance between recently separated populations within archipelagos. We detected SNPs putatively under selection within the Canarian and Madeiran archipelagos, but found no overlap between candidate SNPs identified from previous analyses at a broader spatial scale, i.e. between archipelagos [31]. We found evidence for selection at SNPs associated with head length across the Madeiran islands, and for an SNP located between candidate genes involved in the regulation of DNA repair and enzyme pathways across the Canary Islands.

Previous studies have used microsatellites to examine modes and patterns of population divergence across the three archipelagos colonized by the Berthelot's pipit [29,30], while more recent studies have used genome-scale analyses to examine broad-scale population structure and bottlenecks between archipelagos [31]. Here, we complement these findings by inferring colonization and gene flow at different geographical scales using *TreeMix*, and examine population-level patterns of genetic diversity using LD decay. *TreeMix* shows strong divergence between the tawny and Berthelot's pipit (electronic supplementary material, figure S2A), consistent with phylogeny-based estimates [28]. Tree topology suggests initial divergence of Berthelot's pipit from the tawny pipit may have been to the central or eastern Canary Islands (electronic supplementary material, figure S2A), with long archipelago branch lengths consistent with independent colonization events to the Madeiran and the Selvagens archipelagos with associated bottlenecks (figure 2a). There are weak signatures of within-archipelago divergence and structure ($F_{ST}$ 0.01–0.06), with the longest population branch lengths (figure 2a) and individual PCA clustering (figure 3) across the small isolated populations of El Hierro, La Graciosa and Deserta Grande relative to other within-archipelago populations. Past colonization history and associated bottlenecks are reflected in patterns of population-level LD; rapid LD decay at proximate SNPs and low long-range LD indicates larger and more outbred populations across the Canary Islands, while high long-range LD indicate bottlenecks and/or inbreeding and reduced genetic diversity [25,53]. Our patterns of LD are consistent with reduced genetic diversity in the Madeira and Selvagens archipelagos (figure 2b) and support previously reported patterns based on microsatellite and RAD data (see [29,31]). Simulation-based approaches, using a greater density of SNPs, may be useful to further confirm if our LD patterns are as a result of population bottlenecks or to add detail to the population size estimates at different historical time points. One common feature of population-level LD decay is a dip in the regression between 25 and 50 kb followed by a rise. This pattern has been found in previous studies of LD in this system using the loess line fitting method [31]. We are unable to determine a biological explanation for this, but alternative line fitting methods such as those used by Hill & Weir [59] reflect our archipelago level conclusions.

Limited gene flow between island populations of the Berthelot's pipit has previously been suggested based on strong genetic structure between archipelagos [29–31], while stable host–pathogen communities within populations suggest limited movement between closely located islands [33]. Here, we explicitly test for gene flow between islands and find that adding migration events between populations did not significantly improve our model of population history. Further, high LD and reduced genetic diversity in the Madeiran archipelago and Selvagens are consistent with the absence of contemporary gene flow between archipelagos. These findings are reflected in other studies that investigate admixture between populations diverging at different levels of geographical separation and across differing timescales [60,61]. Given that levels of divergence differ between pairs of closely located Berthelot's pipit populations, we cannot discount the possibility of weak gene flow slowing the accumulation of genetic divergence between some pairs of recently separated populations which we have been unable to detect using our marker set.

We aimed to identify signatures of selection within archipelagos to uncover ecologically relevant adaptation between recently separated island populations of Berthelot's pipit, which may be eroded by other evolutionary processes at broader scales. Applying a reverse genetics approach identified loci with patterns of variation consistent with natural selection across ecological gradients within archipelagos, while controlling for neutral genome-wide divergence owing to structure. The EIGENGWAS analyses detected loci under divergent selection within both the Canary Islands and Madeiran archipelago, with different SNPs detected within each archipelago (table 1). Further, we found no evidence for a correlation between locus $F_{ST}$ values within the two sets of archipelagos, and no markers with high levels of structure within both archipelagos (figure 5). This suggests that either (i) different selective pressures act across the archipelagos resulting in outliers associated with different genes, or (ii) selection acts on similar traits, but these traits vary in genetic architecture among archipelagos. We identified one significant candidate SNP (219s24) as putatively under divergent selection across the Canary Islands. This SNP was in a gene-dense region of chromosome 5, with *WDHD1* upstream and *GCH1* downstream. These genes act in DNA repair [56] and enzyme pathways [57], respectively, but their function in Berthelot's pipits is unknown. It is worth pointing out that while we do not detect selection for the same SNPs within archipelagos as between archipelagos, our outlier SNP in the Canary Islands, 219s24, was in the same broad genomic region on chromosome 5 as a set of SNPs previously identified as being putatively under selection and associated with bill length across the range of Berthelot's pipit [31]. Further research, with a higher density of SNPs, is needed to identify the importance of this genomic region for adaptation in the pipit system. Using our marker set, we detected only four loci under selection in our analyses. We see low levels of genetic variation, and hence few SNPs, within archipelagos of the Berthelot's pipit as a result of colonization history and bottlenecks, but it is also likely that we have no marker coverage in some regions of the genome that may be under selection. Therefore, future studies are needed to uncover greater detail on the loci and hence traits that are under selection in this system.

The most significant SNPs (1585s94 and 1585s112) in the EIGENGWAS analysis of Madeiran populations mapped to the *ADAM12* gene, which has been linked to body growth through skeletal development in zebrafish (*Danio rerio*) and forms part of a family of proteins involved in development, homeostasis and disease [58]. Mixed model analysis, based on all samples across the Berthelot's pipit's range, revealed that the genotype at these loci was significantly associated with head length in this species. There was a low MAF (less than 15%) for both SNPs across the Canary Island populations and Selvagem Grande. In the Madeiran archipelago, the minor allele was absent from both Madeira and Porto Santo, but at 50% prevalence in Deserta Grande (electronic supplementary material, table S3). As discussed, colonization of the Madeiran archipelago was from the Canary Islands, and involved a bottleneck, with a further bottleneck during the subsequent colonization of Deserta Grande. Based on this history, there are two potential explanations as to why we see these genotype patterns across the Madeiran populations. Firstly, positive selection may have increased the frequency of the minor allele on Deserta Grande. Alternatively, the minor allele may have been lost owing to purifying/negative selection, or random genetic drift, on the other Madeiran islands while being maintained on Deserta Grande. This second scenario is unlikely as we see greater diversity at these SNPs in the most bottlenecked population where we would expect the lowest levels of diversity. The genotype–phenotype relationship for the SNP within the *ADAM12* gene was no longer significant when two individuals (out of 19) with particularly large beaks were removed from Deserta Grande, and larger sample sizes are needed to determine the robustness of this result. Nonetheless, this research adds to the growing body of evidence that genes associated with craniofacial development may be excellent candidates for the study of natural selection in wild birds [16,62,63].

In this study, we identified SNPs putatively under selection in recently diverged island populations of Berthelot's pipit within archipelagos. Using the same EIGENGWAS approach, we previously observed a

larger number of divergent selection signatures *between* archipelagos, identifying dozens of SNPs putatively under selection (see [31]). Our findings are similar to those seen in other studies that have used genome scans to investigate adaptation at different spatial scales in the wild. The strength of genetic differentiation and selection increases with geographical distance between Mascarene grey white-eye (*Zosterops borbonicus*; [64]) and barn swallow populations, (*Hirundo rustica*; [65]) and between lake and ocean populations in brown trout (*Salmo trutta*; [66]) and Atlantic salmon (*Salmo salar*; [67]). Adaptive divergence between geographically close populations is expected to be eroded if high gene flow between populations counteracts selection; however, an increasing number of studies show that local adaptation can persist despite gene flow (reviewed in [27,68,69]). Given that strong differences in pathogen prevalence, habitat and climatic conditions exist between closely related Berthelot's pipit populations—and we provide evidence of only very weak gene flow between populations—the low number of outlier SNPs within compared to across archipelagos more likely reflects weak selection between recently separated populations instead of gene flow counteracting selection in this system.

## 5. Conclusion

Combining the study of population history, drift and selection between island populations at different spatial scales provides an opportunity to understand how evolution shapes variation in nature. We assessed contemporary patterns of variation across the range of Berthelot's pipit, revealing that genetic diversity is largely shaped by colonization events, with very weak evidence of gene flow between islands. We uncover outlier loci putatively under divergent selection between recently separated populations within archipelagos. Patterns of diversity at these loci, and the ecological adaptation they may be involved in, may be masked by other evolutionary processes when assessing genetic variation at broader scales. Our findings suggest natural selection may act repeatedly on traits, particularly bill morphology, at different spatial scales, and that signals of selection appear to be weaker between recently separated populations. Moving forward, studying demography and selection at a range of spatial scales is likely to prove a powerful approach for determining the strength and nature of adaptation in the wild.

Ethics. Permission to sample and ring Berthelot's pipits across their range was obtained from the Spanish Environment Ministry, the Canary government and the Natural Park of Madeira.

Data accessibility. All data and code used in this manuscript have been deposited on the Dryad Digital Repository: https://doi.org/10.5061/dryad.pc866t1kt [70]. This includes genomic .bed and associated .bim and .fam files, phenotypic and population information for each pipit sample, and scripts for data processing and production of figures and tables. The .qseq file of raw RAD reads for each individual sample are also available on the Dryad Digital Repository: https://doi.org/10.5061/dryad.9642b.

Authors' contributions. L.G.S. and D.S.R. designed the research and obtained the funding, and with B.C.E., supervised the project; L.G.S. and J.C.I. (and to a lesser extent, D.S.R.) undertook the field sampling; C.A. performed the ddRAD sequencing and bioinformatics while affiliated with NBAF Sheffield; C.A.M. and L.G.S. analysed the data, with input from C.A., and wrote the manuscript. All authors input to, and approved the final version of the manuscript.

Competing interests. We declare we have no competing interests.

Funding. This work was supported by Natural Environment Research Council (NERC) studentships to C.A.M. and C.A. (NE/L002582/1), a Norwich Research Park Science Links grant to D.S.R., M.C. and L.G.S., and a BBSRC fellowship (BB/N011759/1) and British Ecological Society Large Research Grant to L.G.S.

Acknowledgements. We thank two anonymous reviewers whose insight and helpful suggestions have improved this manuscript. We thank Matthew Clark and Lawrence Percival-Alwyn for assistance generating the Berthelot's pipit genome, and Helen Hipperson, Clemens Küpper and Terry Burke (NERC Biomolecular Analysis Facility—Sheffield) for assistance with RAD sequencing. We also thank the Spanish Environment Ministry, the Canary government and the Natural Park of Madeira who gave permission for us to perform the sampling work, local governments in the Canary Islands and Madeira who provided accommodation, and the Portuguese Navy who provided transport. The research presented in this paper was carried out on the High Performance Computing Cluster supported by the Research and Specialist Computing Support service at the University of East Anglia.

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
