## [Peer Review File · Royal Society Open Science]

Review History

RSOS-201146.R0 (Original submission)

Review form: Reviewer 1

Is the manuscript scientifically sound in its present form?

Yes

Are the interpretations and conclusions justified by the results?

Yes

Is the language acceptable?

Yes

Do you have any ethical concerns with this paper?

No

Have you any concerns about statistical analyses in this paper?

No

Recommendation?

Accept with minor revision (please list in comments)

Comments to the Author(s)

I think this is a very interesting paper on genetic variation within archipelago in the Berthelot's pipit, a passerine found in three archipelagos of the North Atlantic (the Canary islands, the Selvagens and the Madeiran archipelago). As far as I can tell, the analyses are well done, except for one major concern that I have regarding the analyses related to linkage disequilibrium (LD) (see major comment (Appendix A)). I mostly have small comments and suggestions to make the manuscript a little bit clearer but this is a very polished version already and I'm recommending to accept this paper with minor revisions.

Review form: Reviewer 2**Is the manuscript scientifically sound in its present form?**

Yes

Are the interpretations and conclusions justified by the results?

No

Is the language acceptable?

Yes

Do you have any ethical concerns with this paper?

No

Have you any concerns about statistical analyses in this paper?

Yes

Recommendation?

Major revision is needed (please make suggestions in comments)

Comments to the Author(s)

In your paper "Genomic variation, population history and within-archipelago adaptation between island bird populations" you perform population genetic analyses and search for loci under divergent selection in Berthelot's pipits sampled across three Macaronesia archipelagos. In general, I enjoyed reading your manuscript although it was at times hard to follow given the many analyses performed and the large supplement. I have a few major points that you may want to consider (order is arbitrary and not by importance):

1. Please provide some more details on the RAD-data, e.g. coverage per individual and software used. I think that these minimal details should be provided here rather than referring to Armstrong et al. (2018).
2. You mention reduced genetic diversity on the smaller islands several times in the manuscript but never estimated π . This should be possible with your RAD-data and would be a nice addition.
3. EigenGWAS: I never used that program but as far as I understand it takes the PC1 scores of each individual as the dependent variable and each SNP (used for the PCA) as the independent

variable. I guess that it does not correct for relatedness.? Thus, it should be similar to the method described in Duforet-Frebourg et al. (2016) or used in Knief et al. (2019). It is basically a representation of the loadings, right? Thus, it (A) is not too surprising that the SNPs you detect are also those with the highest F_{ST} (Fig. 5). (B) I was wondering if in the Madeiran archipelago the two significant SNPs generate PC1 because they are in high LD ($r^2 > 0.982$). Would it not be more informative to use the LD-pruned SNP set here? Also, I was wondering whether a Bonferroni-corrected significant threshold is conservative here given that SNPs are in LD. The method implemented in Gao et al. (2008) may be useful here. Last, throughout the manuscript you write “SNPs under selection” (except for the Conclusions paragraph). I would rather say “SNPs putatively under selection” because EigenGWAS does not proof selection.

4. Genotype-phenotype associations: Why do you not fit your relatedness matrix as a random effect (and use all individuals, including the related ones)? Accounting for population is not enough to control the type I error rate here. You used lme4 for fitting the LMMs (not described in the Methods section though and only visible in the provided code) but I think that with the R-package lme4qtl you should be able to do this (Ziyatdinov et al. 2018; although I have never used that specific package, I know that lme4 can be modified to handle any relatedness matrix: this function was called relmatmm and can be found via Google). I am wondering why you use principal components and not the raw phenotypes in your genotype-phenotype associations. The PCs are hard to interpret (except for PC1, which mostly reflects size) and I would thus suggest that you use the raw phenotypes as dependent variables and – in case you want this – correct for size by fitting PC1 as an independent variable in your model. Also, using all these PCs may look like data dredging.

5. Table S1: Although the supplement is already extensive, I think that you should split that table into two. Table S1A: The part above the dashed line (i.e. which trimming/filtering steps used for which analyses). Table S1B: Put the different populations as columns and provide the total numbers of individuals and SNPs as rows. Add rows for each filtering step and give the number of individuals/SNPs removed (or kept). Consider adding these two tables to the main text.

6. LD-analysis: Please provide details on the local regression model used for fitting the regression lines in the Methods section. In Fig. 2 I was wondering why there is always a “dip” in the regression line between 25 and 50 kb. Does the local regression model provide estimates such that you can statistically show that “baseline” LD is higher in populations on smaller islands (some kind of intercept)? Why do you not show all populations in Fig. 2 (rather than having an additional Fig. S7)? I think that adding the Selvagens to Fig. 2 is necessary.

7. TreeMix: I am not very familiar with this software but to me the branches in Fig. 2A and even more so in Fig. S2A appear very short. However, you base some conclusions (i.e. that TF was colonized first) partly on the branch lengths. Is this justified?

8. Provide line numbers for the next revision. My minor comments are just “approximate” line numbers.

I hope you find these comments helpful and best wishes.

Minor comments:

P3L5: “evolutionary processes”

P3L13 and throughout: Could you reformulate “very limited evidence”? Do you have little evidence (i.e. there may or may not be gene flow) or little (if any) gene flow?

P3L32: Why not use “depends”?

P3L38: “Studies on humans”

P3L39: “important for our understanding”

P3L41: “insights”

P4L13: please reformulate “evolutionary significant adaptations”. Significant is a word used in statistics. Are there also “evolutionary non-significant adaptations? Likewise “evolutionary important adaptations” (L21) sounds strange.

P4L14: replace “subsequent” with “other”

P4L57: So it is a question of parallelism at the molecular level? Do birds from different archipelagos from similar habitats resemble each other?

P5L13: You do not quantify selection (differentials). So you cannot address the question of “how strong”.

P5L14: You have only a handful of (morphological) traits.

P5L46: Remove “also”

P5L54: I do not understand what is meant by “fewer than 4 ambiguous genotypes”.

P5L58: “start (s)”

P6L14: “included in the”. You should maybe write here how many were removed.

P7L12: How sensitive are your analyses to these thresholds?

P7L22: Refer to Tab. S1 here.

P7L50 and throughout: Check spelling of “principal component analysis”

P7L53: “close relatives and SNPs with a MAF < 0.03 were removed”

P7L54: “filter based on LD”

P7L60: Maybe briefly describe what EigenGWAS does.

P8L21: Which software did you use for fitting LMMs?

P8L25: Replace “run” with “fitted”

Fig. 2 and throughout: You could color the populations stemming from the same archipelago and use different symbols but the same color for populations within archipelagos.

P9L30-32: I do not see this in Tab. S2. Also, these are a lot of tests.

P9L36: “Thus, across all populations”

P9L38: How do you define “baseline”

P10L28: Are these SNPs located in the coding region? Synonymous?

P13L4: I cannot see the shaded bands

P15L48: “ $P < 1.70 \times 10^{-5}$ ”

Fig. 5: Think about moving it to the supplement

P16L30: Where does the number “3531” come from? Why not all 4470 SNPs?

P16L39: “ADAM12 has been” ... “is therefore a potential”

P16L49: Based on what did you decide showing results from SNP1585s112?

P16L52+55: I think it is “Fig. 11”

P16L52: “Homozygous”

P17L39: From the Canary Islands, not from the mainland

P17L45: “larger and more”

P17L46: “islands, and increasingly”

P17L47: remove “through bottlenecks and” (not shown)

P17L56: is it “strong” or “high” LD. Replace “of” with “and”

P18L1-7: Maybe remove?

P18L37: “that the genotype”

References: standardize lower and upper cases

P27L15: Why did you use windows of 100 SNPs and not some distance measure?

P27L16: “but was dominated”

Tab. S2: Not ordered by Z-scores. Could you make this more readable, e.g. by ordering the triplets alphabetically? In these analyses, do you need to account for multiple testing?

Tab. S4: Provide variance explained by the PCs. “Principal component”

Fig. S2B: Could you explain the high values between TAW, M, PS and DG?

References

- Armstrong C, Richardson DS, Hipperson H, Horsburgh GJ, Kupper C, Percival-Alwyn L, . . . Spurgin LG (2018) Genomic associations with bill length and disease reveal drift and selection across island bird populations. *Evolution Letters* 2, 22–36.
- Duforet-Frebourg N, Luu K, Laval G, Bazin E, Blum MG (2016) Detecting genomic signatures of natural selection with principal component analysis: application to the 1000 genomes data. *Molecular biology and evolution* 33, 1082–1093.
- Gao XY, Stamier J, Martin ER (2008) A multiple testing correction method for genetic association studies using correlated single nucleotide polymorphisms. *Genet Epidemiol* 32, 361–369.
- Knief U, Bossu CM, Saino N, Hansson B, Poelstra J, Vijay N, . . . Wolf JBW (2019) Epistatic mutations under divergent selection govern phenotypic variation in the crow hybrid zone. *Nature ecology & evolution* 3, 570–576.
- Ziyatdinov A, Vazquez-Santiago M, Brunel H, Martinez-Perez A, Aschard H, Soria JM (2018) lme4qtl: linear mixed models with flexible covariance structure for genetic studies of related individuals. *BMC bioinformatics* 19, e68.

Decision letter (RSOS-201146.R0)

Dear Ms Martin

The Editors assigned to your paper RSOS-201146 "Genomic variation, population history and within-archipelago adaptation between island bird populations" have now received comments from reviewers and would like you to revise the paper in accordance with the reviewer comments and any comments from the Editors. Please note this decision does not guarantee eventual acceptance.

Please submit your revised manuscript and required files (see below) no later than 21 days from today's (ie 29-Sep-2020) date. Note: the ScholarOne system will 'lock' if submission of the revision is attempted 21 or more days after the deadline. If you do not think you will be able to meet this deadline please contact the editorial office immediately.

on behalf of the Associate Editor, and Professor Kevin Padian (Subject Editor)
openscience@royalsociety.org

Associate Editor Comments to Author:

Thank you for the submission. We'd like you to revise the paper to take into account the reviewers queries and comments.

Reviewer comments to Author:

Reviewer: 1

Comments to the Author(s)

I think this is a very interesting paper on genetic variation within archipelago in the Berthelot's pipit, a passerine found in three archipelagos of the North Atlantic (the Canary islands, the Selvagens and the Madeiran archipelago). As far as I can tell, the analyses are well done, except for one major concern that I have regarding the analyses related to linkage disequilibrium (LD) (see major comment). I mostly have small comments and suggestions to make the manuscript a little bit clearer but this is a very polished version already and I'm recommending to accept this paper with minor revisions.

Reviewer: 2

Comments to the Author(s)

In your paper "Genomic variation, population history and within-archipelago adaptation between island bird populations" you perform population genetic analyses and search for loci under divergent selection in Berthelot's pipits sampled across three Macaronesia archipelagos. In general, I enjoyed reading your manuscript although it was at times hard to follow given the many analyses performed and the large supplement. I have a few major points that you may want to consider (order is arbitrary and not by importance):

1. Please provide some more details on the RAD-data, e.g. coverage per individual and software used. I think that these minimal details should be provided here rather than referring to Armstrong et al. (2018).
2. You mention reduced genetic diversity on the smaller islands several times in the manuscript but never estimated π . This should be possible with your RAD-data and would be a nice addition.
3. EigenGWAS: I never used that program but as far as I understand it takes the PC1 scores of each individual as the dependent variable and each SNP (used for the PCA) as the independent variable. I guess that it does not correct for relatedness.? Thus, it should be similar to the method described in Duforet-Frebourg et al. (2016) or used in Knief et al. (2019). It is basically a representation of the loadings, right? Thus, it (A) is not too surprising that the SNPs you detect are also those with the highest F_{ST} (Fig. 5). (B) I was wondering if in the Madeiran archipelago

the two significant SNPs generate PC1 because they are in high LD ($r^2 > 0.982$). Would it not be more informative to use the LD-pruned SNP set here? Also, I was wondering whether a Bonferroni-corrected significant threshold is conservative here given that SNPs are in LD. The method implemented in Gao et al. (2008) may be useful here. Last, throughout the manuscript you write “SNPs under selection” (except for the Conclusions paragraph). I would rather say “SNPs putatively under selection” because EigenGWAS does not proof selection.

4. Genotype-phenotype associations: Why do you not fit your relatedness matrix as a random effect (and use all individuals, including the related ones)? Accounting for population is not enough to control the type I error rate here. You used lme4 for fitting the LMMs (not described in the Methods section though and only visible in the provided code) but I think that with the R-package lme4qtl you should be able to do this (Ziyatdinov et al. 2018; although I have never used that specific package, I know that lme4 can be modified to handle any relatedness matrix: this function was called relmatmm and can be found via Google). I am wondering why you use principal components and not the raw phenotypes in your genotype-phenotype associations. The PCs are hard to interpret (except for PC1, which mostly reflects size) and I would thus suggest that you use the raw phenotypes as dependent variables and – in case you want this – correct for size by fitting PC1 as an independent variable in your model. Also, using all these PCs may look like data dredging.

5. Table S1: Although the supplement is already extensive, I think that you should split that table into two. Table S1A: The part above the dashed line (i.e. which trimming/filtering steps used for which analyses). Table S1B: Put the different populations as columns and provide the total numbers of individuals and SNPs as rows. Add rows for each filtering step and give the number of individuals/SNPs removed (or kept). Consider adding these two tables to the main text.

6. LD-analysis: Please provide details on the local regression model used for fitting the regression lines in the Methods section. In Fig. 2 I was wondering why there is always a “dip” in the regression line between 25 and 50 kb. Does the local regression model provide estimates such that you can statistically show that “baseline” LD is higher in populations on smaller islands (some kind of intercept)? Why do you not show all populations in Fig. 2 (rather than having an additional Fig. S7)? I think that adding the Selvagens to Fig. 2 is necessary.

7. TreeMix: I am not very familiar with this software but to me the branches in Fig. 2A and even more so in Fig. S2A appear very short. However, you base some conclusions (i.e. that TF was colonized first) partly on the branch lengths. Is this justified?

8. Provide line numbers for the next revision. My minor comments are just “approximate” line numbers.

I hope you find these comments helpful and best wishes.

Minor comments:

P3L5: “evolutionary processes”

P3L13 and throughout: Could you reformulate “very limited evidence”? Do you have little evidence (i.e. there may or may not be gene flow) or little (if any) gene flow?

P3L32: Why not use “depends”?

P3L38: “Studies on humans”

P3L39: “important for our understanding”

P3L41: “insights”

P4L13: please reformulate “evolutionary significant adaptations”. Significant is a word used in statistics. Are there also “evolutionary non-significant adaptations”? Likewise “evolutionary important adaptations” (L21) sounds strange.

P4L14: replace “subsequent” with “other”

P4L57: So it is a question of parallelism at the molecular level? Do birds from different archipelagos from similar habitats resemble each other?

P5L13: You do not quantify selection (differentials). So you cannot address the question of “how strong”.

P5L14: You have only a handful of (morphological) traits.

P5L46: Remove “also”

P5L54: I do not understand what is meant by “fewer than 4 ambiguous genotypes”.

P5L58: “start (s)”

P6L14: “included in the”. You should maybe write here how many were removed.

P7L12: How sensitive are your analyses to these thresholds?

P7L22: Refer to Tab. S1 here.

P7L50 and throughout: Check spelling of “principal component analysis”

P7L53: “close relatives and SNPs with a MAF < 0.03 were removed”

P7L54: “filter based on LD”

P7L60: Maybe briefly describe what EigenGWAS does.

P8L21: Which software did you use for fitting LMMs?

P8L25: Replace “run” with “fitted”

Fig. 2 and throughout: You could color the populations stemming from the same archipelago and use different symbols but the same color for populations within archipelagos.

P9L30-32: I do not see this in Tab. S2. Also, these are a lot of tests.

P9L36: “Thus, across all populations”

P9L38: How do you define “baseline”

P10L28: Are these SNPs located in the coding region? Synonymous?

P13L4: I cannot see the shaded bands

P15L48: “ $P < 1.70 \times 10^{-5}$ ”

Fig. 5: Think about moving it to the supplement

P16L30: Where does the number “3531” come from? Why not all 4470 SNPs?

P16L39: “ADAM12 has been” ... “is therefore a potential”

P16L49: Based on what did you decide showing results from SNP1585s112?

P16L52+55: I think it is “Fig. 11”

P16L52: “Homozygous”

P17L39: From the Canary Islands, not from the mainland

P17L45: “larger and more”

P17L46: “islands, and increasingly”

P17L47: remove “through bottlenecks and” (not shown)

P17L56: is it “strong” or “high” LD. Replace “of” with “and”

P18L1-7: Maybe remove?

P18L37: “that the genotype”

References: standardize lower and upper cases

P27L15: Why did you use windows of 100 SNPs and not some distance measure?

P27L16: “but was dominated”

Tab. S2: Not ordered by Z-scores. Could you make this more readable, e.g. by ordering the triplets alphabetically? In these analyses, do you need to account for multiple testing?

Tab. S4: Provide variance explained by the PCs. “Principal component”

Fig. S2B: Could you explain the high values between TAW, M, PS and DG?

References

Armstrong C, Richardson DS, Hipperson H, Horsburgh GJ, Kupper C, Percival-Alwyn L, . . . Spurgin LG (2018) Genomic associations with bill length and disease reveal drift and selection across island bird populations. *Evolution Letters* 2, 22–36.

Duforet-Frebourg N, Luu K, Laval G, Bazin E, Blum MG (2016) Detecting genomic signatures of natural selection with principal component analysis: application to the 1000 genomes data. *Molecular biology and evolution* 33, 1082–1093.

Gao XY, Stamier J, Martin ER (2008) A multiple testing correction method for genetic association studies using correlated single nucleotide polymorphisms. *Genet Epidemiol* 32, 361–369.

Knief U, Bossu CM, Saino N, Hansson B, Poelstra J, Vijay N, . . . Wolf JBW (2019) Epistatic mutations under divergent selection govern phenotypic variation in the crow hybrid zone. *Nature ecology & evolution* 3, 570–576.

Ziyatdinov A, Vazquez-Santiago M, Brunel H, Martinez-Perez A, Aschard H, Soria JM (2018) lme4qtl: linear mixed models with flexible covariance structure for genetic studies of related individuals. *BMC bioinformatics* 19, e68.

===PREPARING YOUR MANUSCRIPT===

Your revised paper should include the changes requested by the referees and Editors of your manuscript. You should provide two versions of this manuscript and both versions must be provided in an editable format:
 one version identifying all the changes that have been made (for instance, in coloured highlight, in bold text, or tracked changes);
 a 'clean' version of the new manuscript that incorporates the changes made, but does not highlight them. This version will be used for typesetting if your manuscript is accepted.
 Please ensure that any equations included in the paper are editable text and not embedded images.

===PREPARING YOUR REVISION IN SCHOLARONE===

Author's Response to Decision Letter for (RSOS-201146.R0)

See Appendix B.

RSOS-201146.R1 (Revision)

Review form: Reviewer 1

Is the manuscript scientifically sound in its present form?

Yes

Are the interpretations and conclusions justified by the results?

Yes

Is the language acceptable?

Yes

Do you have any ethical concerns with this paper?

No

Have you any concerns about statistical analyses in this paper?

No

Recommendation?

Accept with minor revision (please list in comments)

Comments to the Author(s)

Please see attached file (Appendix C).

Review form: Reviewer 2

Is the manuscript scientifically sound in its present form?

Yes

Are the interpretations and conclusions justified by the results?

Yes

Is the language acceptable?

Yes

Do you have any ethical concerns with this paper?

No

Have you any concerns about statistical analyses in this paper?

No

Recommendation?

Accept with minor revision (please list in comments)

Comments to the Author(s)

I acted as reviewer # 2 for the initial submission of your manuscript "Genomic variation, population history and within-archipelago adaptation between island bird populations". I think

you now improved the manuscript considerably, taking also my comments and suggestions into account. I have only one major comment remaining and some minor edits:

1. LD-based inferences on N_e . I do not think that `loess()` is the most appropriate way of fitting a LD-decay function to your data. In the end, you do not get any parameters or statistics on which you could base your conclusions. Also, (1) how did you define the span-parameter (and based on what?) and (2) `loess()` is known to overreact which may explain the dip around 25–50 kb distance. Thus, I would suggest using an appropriate model here, as for example done in Knief et al. (2017): You could fit the function described in Hill & Weir (1988), which is

$$LD.data \sim ((10 + \rho * distance) / ((2 + \rho * distance) * (11 + \rho * distance))) * (1 + ((3 + \rho * distance) * (12 + 12 * \rho * distance + (\rho * distance)^2)) / (n * (2 + \rho * distance) * (11 + \rho * distance)))$$

where n is the sample size and ρ represents the population recombination parameter ($\rho = 4 * N_e * r$) that is going to be estimated.

Given your LD data in a data frame called “out” with columns “Distance” and “Pearson_r2”, you can use the following R code:

```
out1 <- out[order(out$Distance),]
distance <- out1$Distance
LD.data <- out1$Pearson_r2
n <- 2*939
HW.st <- c(C=0.1)
HW.nonlinear <-
nls(LD.data ~ ((10+C*distance) / ((2+C*distance)*(11+C*distance))) * (1 + ((3+C*distance)*(12+12*C*distance + (C*distance)^2)) / (n*(2+C*distance)*(11+C*distance))), start=HW.st, control=nls.control(maxiter=100))
tt <- summary(HW.nonlinear)
new.rho <- tt$parameters[1]
fpoints <-
((10+new.rho*distance) / ((2+new.rho*distance)*(11+new.rho*distance))) * (1 + ((3+new.rho*distance)*(12+12*new.rho*distance + (new.rho*distance)^2)) / (n*(2+new.rho*distance)*(11+new.rho*distance)))
```

and plot the estimated curve with

```
points(distance, fpoints, col="orangered", type="l")
```

You could then compare your estimates of the parameter “C”, e.g. by getting confidence intervals with

```
confint(HW.nonlinear)
```

I guess (but have never done so) that you could also get bootstrapped confidence bands for the fitted lines (see, for example, <https://cran.r-project.org/web/packages/nlraa/vignettes/Bootstrapping.html>).

You could also estimate the LD-half-decay, e.g. through:

```
# Choose between actual and estimated values:
```

```
HalfDecay <- max(LD.data)/2
```

```
HalfDecay <- max(fpoints)/2
```

```
down <- out1[max(which(fpoints>=HalfDecay)), ]
up <- out1[min(which(fpoints<HalfDecay)), ]
half.decay.distance <- out1[min(which(abs(fpoints-HalfDecay)), ]
```

I think that this formula (and others) are also implemented in ngsLD (Fox et al. 2019).

I hope you find these comments helpful and best wishes.

Minor comments (line numbers refer to the document with track changes):

L84: "selection can also be"

L87: "may reveal patterns of strong"

L95: "adaptations"

L108: "including diseases and climatic factors"

L132: Do you mean divergent or disruptive selection instead of "differential"?

L141: "across the species range"

L151: "variation to variation in phenotypic"

L173: "and birds were molecularly"

L223: "population structure"

L228: Here and throughout: Check the tense you are writing in. I think it must be simple past here, i.e. "We implemented"

L253: "analyses"

L265: Here and throughout: Do you mean "tree topology" instead of "tree topography"?

L265: "pilot populations, we computed"

L281: "larger, more outbred populations show a rapid"

L288: "Ne, and LD at"

L290: "calculated" or "estimated"

L294: "the R function". How did you set the span parameter?

L305: "LD, which enabled us"

L306: "implemented in"

L309: "individually for an association with these axes"

L313: I guess you either mean above the $-\log_{10}(P)$ threshold or below the P-value threshold.

L318-320: Delete that sentence, it is repetitive with the above.

L325: "After having"

L329: Please add version numbers

L329: "implemented in R"

L332: "SNPs, we fitted LMMs [...] as dependent variables."

L332: Why do you not test all other phenotypic traits, e.g. wing length?

L347: "tree, we trimmed"

L366: "showed"

L375: "explained" and "variance explained"

L378: "placed"

L388: "especially in the central"

L389: "highest" instead of "strongest"

L401: "suggested few"

L402: "occurred" instead of "exist"?

L407: "smallest and most isolated"

L411: You could add the half-decay distances here.

L455: "6,252 bases downstream"

L492: I do not think that "Analysis of the first" is correct here. It is just the plot of the 1st PC vs 2nd PC.

L537: "genotype associations"

L538: If you do not use all phenotypes then delete them from the methods section. However, I strongly encourage you to present all results, including those for wing length and weight.

L546: "putatively under selection"

L577: I think it was "head length", not "beak length".?

L578: "and for a SNP in a candidate gene"

L587: This needs a quantitative assessment, as described in my major point above.

L598: "larger and more outbred"

L605: What are these previous studies?

L640: "putatively under divergent"

L654: "may be under selection"

L662-663: I think this is not correct. Or did you control for size somehow?

L668: "history, there"

L682: "under selection in recently diverged"

L734: Please provide also the doi/accession number for the raw sequencing data

References

Fox EA, Wright AE, Fumagalli M, Vieira FG (2019) ngsLD: evaluating linkage disequilibrium using genotype likelihoods. *Bioinformatics* 35, 3855-3856.

Hill WG, Weir BS (1988) Variances and covariances of squared linkage disequilibria in finite populations. *Theor Popul Biol* 33, 54-78.

Knief U, Schielzeth H, Backström N, Hemmrich-Stanisak G, Wittig M, Franke A, . . . Forstmeier W (2017) Association mapping of morphological traits in wild and captive zebra finches: reliable within, but not between populations. *Mol Ecol* 26, 1285-1305.

Decision letter (RSOS-201146.R1)

Dear Miss Martin

On behalf of the Editors, we are pleased to inform you that your Manuscript RSOS-201146.R1 "Genomic variation, population history and within-archipelago adaptation between island bird populations" has been accepted for publication in Royal Society Open Science subject to minor revision in accordance with the referees' reports. Please find the referees' comments along with any feedback from the Editors below my signature.

Please submit your revised manuscript and required files (see below) no later than 7 days from today's (ie 14-Dec-2020) date. Note: the ScholarOne system will 'lock' if submission of the revision is attempted 7 or more days after the deadline. If you do not think you will be able to meet this deadline please contact the editorial office immediately.

Please note article processing charges apply to papers accepted for publication in Royal Society Open Science (<https://royalsocietypublishing.org/rsos/charges>). Charges will also apply to papers transferred to the journal from other Royal Society Publishing journals, as well as papers

submitted as part of our collaboration with the Royal Society of Chemistry (<https://royalsocietypublishing.org/rsos/chemistry>). Fee waivers are available but must be requested when you submit your revision (<https://royalsocietypublishing.org/rsos/waivers>).

on behalf of Prof Kevin Padian (Subject Editor)
openscience@royalsociety.org

Associate Editor Comments to Author:

The reviewers are much happier with this iteration of your work and we are grateful that you've carefully engaged with their concerns. That said, there are a number of matters that need addressing before your paper may be considered ready for acceptance. Firstly, there are a couple of relatively major issues you'll need to tackle, along with a range of smaller tweaks (a certain amount of copy-editing, for instance). Secondly, it has been observed that raw sequencing reads do not appear to be available through your Dryad deposition - please ensure that you check this and either make clearer how they are accessible in Dryad, or make them available through an additional alternative repository. Good luck!

Reviewer comments to Author:

Reviewer: 2

Comments to the Author(s)

I acted as reviewer # 2 for the initial submission of your manuscript "Genomic variation, population history and within-archipelago adaptation between island bird populations". I think you now improved the manuscript considerably, taking also my comments and suggestions into account. I have only one major comment remaining and some minor edits:

1. LD-based inferences on N_e . I do not think that `loess()` is the most appropriate way of fitting a LD-decay function to your data. In the end, you do not get any parameters or statistics on which you could base your conclusions. Also, (1) how did you define the span-parameter (and based on what?) and (2) `loess()` is known to overreact which may explain the dip around 25–50 kb distance. Thus, I would suggest using an appropriate model here, as for example done in Knief et al. (2017): You could fit the function described in Hill & Weir (1988), which is

$$LD_{.data} \sim ((10 + \rho * distance) / ((2 + \rho * distance) * (11 + \rho * distance))) * (1 + ((3 + \rho * distance) * (12 + 12 * \rho * distance + (\rho * distance)^2)) / (n * (2 + \rho * distance) * (11 + \rho * distance)))$$

where n is the sample size and ρ represents the population recombination parameter ($\rho = 4 * N_e * r$) that is going to be estimated.

Given your LD data in a data frame called "out" with columns "Distance" and "Pearson_r2", you can use the following R code:

```

out1 <- out[order(out$Distance),]
distance <- out1$Distance
LD.data<- out1$Pearson_r2
n <- 2*939
HW.st<-c(C=0.1)
HW.nonlinear <-
nls(LD.data~((10+C*distance)/((2+C*distance)*(11+C*distance)))*(1+((3+C*distance)*(12+12*C*di
stance+(C*distance)^2))/(n*(2+C*distance)*(11+C*distance))),start=HW.st,control=nls.control(ma
xiter=100))
tt <- summary(HW.nonlinear)
new.rho <- tt$parameters[1]
fpoints <-
((10+new.rho*distance)/((2+new.rho*distance)*(11+new.rho*distance)))*(1+((3+new.rho*distance
)*(12+12*new.rho*distance+(new.rho*distance)^2))/(n*(2+new.rho*distance)*(11+new.rho*distan
ce)))

```

and plot the estimated curve with

```
points(distance,fpoints,col="orangered",type="l")
```

You could then compare your estimates of the parameter “C”, e.g. by getting confidence intervals with

```
confint(HW.nonlinear)
```

I guess (but have never done so) that you could also get bootstrapped confidence bands for the fitted lines (see, for example, <https://cran.r-project.org/web/packages/nlraa/vignettes/Bootstrapping.html>).

You could also estimate the LD-half-decay, e.g. through:

```
# Choose between actual and estimated values:
```

```
HalfDecay <- max(LD.data)/2
```

```
HalfDecay <- max(fpoints)/2
```

```
down <- out1[max(which(fpoints>=HalfDecay)), ]
```

```
up <- out1[min(which(fpointshalf.decay.distance <- out1[min(which(abs(fpoints-HalfDecay)), ]
```

I think that this formula (and others) are also implemented in ngsLD (Fox et al. 2019).

I hope you find these comments helpful and best wishes.

Minor comments (line numbers refer to the document with track changes):

L84: “selection can also be”

L87: “may reveal patterns of strong”

L95: “adaptations”

L108: “including diseases and climatic factors”

L132: Do you mean divergent or disruptive selection instead of “differential”?

L141: “across the species range”

L151: “variation to variation in phenotypic”

L173: “and birds were molecularly”

L223: “population structure”

L228: Here and throughout: Check the tense you are writing in. I think it must be simple past here, i.e. “We implemented”

- L253: "analyses"
- L265: Here and throughout: Do you mean "tree topology" instead of "tree topography"?
- L265: "pipit populations, we computed"
- L281: "larger, more outbred populations show a rapid"
- L288: "Ne, and LD at"
- L290: "calculated" or "estimated"
- L294: "the R function". How did you set the span parameter?
- L305: "LD, which enabled us"
- L306: "implemented in"
- L309: "individually for an association with these axes"
- L313: I guess you either mean above the $-\log_{10}(P)$ threshold or below the P-value threshold.
- L318-320: Delete that sentence, it is repetitive with the above.
- L325: "After having"
- L329: Please add version numbers
- L329: "implemented in R"
- L332: "SNPs, we fitted LMMs [...] as dependent variables."
- L332: Why do you not test all other phenotypic traits, e.g. wing length?
- L347: "tree, we trimmed"
- L366: "showed"
- L375: "explained" and "variance explained"
- L378: "placed"
- L388: "especially in the central"
- L389: "highest" instead of "strongest"
- L401: "suggested few"
- L402: "occurred" instead of "exist"?
- L407: "smallest and most isolated"
- L411: You could add the half-decay distances here.
- L455: "6,252 bases downstream"
- L492: I do not think that "Analysis of the first" is correct here. It is just the plot of the 1st PC vs 2nd PC.
- L537: "genotype associations"
- L538: If you do not use all phenotypes then delete them from the methods section. However, I strongly encourage you to present all results, including those for wing length and weight.
- L546: "putatively under selection"
- L577: I think it was "head length", not "beak length".?
- L578: "and for a SNP in a candidate gene"
- L587: This needs a quantitative assessment, as described in my major point above.
- L598: "larger and more outbred"
- L605: What are these previous studies?
- L640: "putatively under divergent"
- L654: "may be under selection"
- L662-663: I think this is not correct. Or did you control for size somehow?
- L668: "history, there"
- L682: "under selection in recently diverged"
- L734: Please provide also the doi/accession number for the raw sequencing data

References

- Fox EA, Wright AE, Fumagalli M, Vieira FG (2019) ngsLD: evaluating linkage disequilibrium using genotype likelihoods. *Bioinformatics* 35, 3855-3856.
- Hill WG, Weir BS (1988) Variances and covariances of squared linkage disequilibria in finite populations. *Theor Popul Biol* 33, 54-78.

Knief U, Schielzeth H, Backström N, Hemmrich-Stanisak G, Wittig M, Franke A, . . . Forstmeier W (2017) Association mapping of morphological traits in wild and captive zebra finches: reliable within, but not between populations. *Mol Ecol* 26, 1285–1305.

Reviewer: 1

Comments to the Author(s)

Please see attached file

===PREPARING YOUR MANUSCRIPT===

Your revised paper should include the changes requested by the referees and Editors of your manuscript. You should provide two versions of this manuscript and both versions must be provided in an editable format: one version identifying all the changes that have been made (for instance, in coloured highlight, in bold text, or tracked changes); a 'clean' version of the new manuscript that incorporates the changes made, but does not highlight them. This version will be used for typesetting.

===PREPARING YOUR REVISION IN SCHOLARONE===

<https://royalsociety.org/journals/authors/author-guidelines/#supplementary-material> to include a suitable title and informative caption. An example of appropriate titling and captioning may be found at https://figshare.com/articles/Table_S2_from_Is_there_a_trade-off_between_peak_performance_and_performance_breadth_across_temperatures_for_aerobic_sc_ope_in_teleost_fishes_/3843624.

Author's Response to Decision Letter for (RSOS-201146.R1)

See Appendix D.

Decision letter (RSOS-201146.R2)

Dear Miss Martin,

It is a pleasure to accept your manuscript entitled "Genomic variation, population history and within-archipelago adaptation between island bird populations" in its current form for publication in Royal Society Open Science.

on behalf of Prof Kevin Padian (Subject Editor)
openscience@royalsociety.org

Appendix A

Comments to the Author(s):

I think this is a very interesting paper on genetic variation within archipelago in the Berthelot's pipit, a passerine found in three archipelagos of the North Atlantic (the Canary islands, the Selvagens and the Madeiran archipelago). As far as I can tell, the analyses are well done, except for one major concern that I have regarding the analyses related to linkage disequilibrium (LD) (see major comment). I mostly have small comments and suggestions to make the manuscript a little bit clearer but this is a very polished version already and I'm recommending to accept this paper with minor revisions.

Major comments:

I was not convinced by the use of LD analyses to infer demography. This requires more explanations and references for the reader to understand how to interpret the LD - genetic distance plots in terms of demography, namely bottlenecks and founder events. Here are the sentences that would need more explanations and references:

P7.L45: "To further understand population demography, including bottlenecks and population size, LD was calculated for each population in Plink"

P17.L34: "examine population level patterns of bottlenecks using linkage disequilibrium (LD) decay."

P17.L44: "Past colonisation history and associated bottlenecks are reflected in patterns of population level LD" please add references and explanations to help us understand the rest of the sentence "these indicate larger more outbred populations across the Canary Islands, but increasingly stronger signatures of inbreeding and reduced genetic diversity through bottlenecks and on small isolated islands".

I was not convinced by your interpretations in terms of founder events and bottlenecks, in my opinion, this is not shown by your analyses:

P17.L17: "Patterns of genetic diversity are consistent with signatures of founder events and geographic isolation."

P19.L30: "revealing that genetic diversity is largely shaped by colonisation events and related bottlenecks"

Here is an additional comment on the LD plots:

P9.L34: Could you provide a quantification about the shallowness of the curve? What is the relationship between shallowness and demography (population size)?

I still think it is really interesting to link LD patterns with demography, and I am aware this has been done (see for instance the review by Waples et al. 2016: Waples, R.K., Larson, W.A., and Waples, R.S. (2016). Estimating contemporary effective population size in non-model species using linkage disequilibrium across thousands of loci. *Heredity* 117, 233–240.) but I was not convinced by the LD plots without any models or curve fitting.

Minor comments:

P5.L15: "To address these questions, we first use demographic analyses to examine colonisation history and gene flow across the species range" I am not sure that Treemix and the LD plots qualify as demographic analyses.

P6.L4: How did you identify sex-linked loci?

P6.L5: Do you have missing data or did you do any filtering based on missing data?

P6.L60: supplementary methods – typo last sentence “by was dominated”

P7.L7: Why do you think there could be gene flow between islands? This would help to justify the use of Treemix.

P7.L20 “within population”: it would help to remind what your populations are, *i.e.* one population per island except in Tenerife with two populations in the lowlands and in the highlands.

P7.L22 “rejecting migration events when the following event did not significantly increase the likelihood of the model”: what number would you consider to be a significant increase of the likelihood?

P7.L25: I was not convinced by the choice of the number of SNPs in one block, I find the number 45 to be quite specific. Also, the results with a block of 50 SNPs (and 50 seems to be very close to 45) are quite different for the Canary populations *i.e.* no east-west trend as discussed in the results and discussion.

P7.L39: typo “Berthelot’s”

P7.L44: “PCA=within archipelago; LD=within populations”. Please clarify this sentence.

P8.L4: What is the general principle of EigenGWAS? What is the measure calculated and how does this relate to selection? What is the link with principal components mentioned in the results?

P8.L19: Please indicate the program that you used for the morphological PCA.

P8.L33: 9,960 SNPs without any filtration seems to be very low. Did you filter by missing data?

P8.L55: I would have liked to see the results of the PCA first before Treemix analyses as it is an exploratory analysis without any model and assumptions.

P9.L12: “long branch length”, I don’t think this branch is that long compared to the very long branches of the Selvagens and Madeiran archipelagos.

P9.L30: “F3 results suggest weak admixture events subsequent to branch divergence may exist between Madeira and Porto Santo and Fuerteventura/Lanzarote and La Graciosa.” Maybe you could add that this is consistent with the geographic distances between the islands. It’s reassuring that you did not infer some possibility of admixture between very distant islands.

P9.L41: How about presenting a PCA with all three archipelagos together?

P9.L45: I would rather say “PCA of the Madeiran archipelago separated Deserta Grande from Madeira and Porto Santo along the first axis, and separated Porto Santo from Deserta Grande on the second axis, with a weak gradient from Madeira to Deserta Grande to Porto Santo (Fig. 3B).”

P9.L57: What is the link between EigenGWAS and the PCs? Please explain this in the methods as well to make the results more understandable.

P10.L24: How close to the SNP 219s24 are the genes *WDHI* and *GCHI*? Please indicate the chromosome here as well.

P12. Figure2A: I would suggest to add colours or bars with colours for archipelagos grouping the three islands of Madeira and with eastern and western and Tenerife groups of the Canaries so that we can more easily read the figure and relate it with the text.

Figure2B: I don’t see the link between LD and Treemix analyses so I would not put them in the same figure. Careful, the “shaded band indicating 95% confidence intervals” does not appear on the plots.

P15. Figure 4: The Z chromosome is not present here? If not, could you indicate why? What about the chromosomes 1A and 4A? When reading the scale ($-\log_{10}(p)$), it is not clear what quantity is calculated, and this should be explained in the methods (see previous comment on EigenGWAS).

P16.L39: Typos “Because ADAM12 has been shown to play a role” and “is therefore a potential candidate”

P16.L42: Typo “A principal component analysis”

P16.L43: Please show the PCA variable correlation plots to support the relationships between the PCs and the variables mentioned in the text and show the morphological PCA in the supplementary materials.

P16.L48: Are the results the same for the other SNP?

P17.L22: I may have missed something but it seems to me that the sentences “no overlap between candidate SNPs identified from previous analyses at a broader spatial scale i.e. between archipelagos” and p18.L25 “It is worth pointing out that our outlier SNP in the Canary Islands, 219s24, was in the same broad genomic region on chromosome 5 as a set of SNPs previously identified as being under selection and associated with bill length across the range of Berthelot’s pipit (Armstrong et al., 2018)” are conflicting.

P17.L37: “Tree topography suggests initial divergence of Berthelot’s pipit from the tawny pipit may have been to the central or eastern Canary Islands (Fig. S2A)”. I don’t think we can say that from figure S2A as the branches of the Canary form a polytomy. I don’t think you can say that from figure 2A as well as from what I understood it is not rooted.

P18.L8: Please check all the “within” to decide with the italic style.

P18.L29: “Further research, with a higher density of SNPs, is needed to identify the importance of this genomic region for adaptation in the pipit system.” Yes I agree! I think you have a relatively low number of SNPs to perform genomic scans compared to what can be done now from whole genome sequences, so it’s important to mention that. You may also elaborate on the very few number of outlier loci that you detected. Do you think you could have missed important outlier loci compared to if you had used whole genome sequences?

P19.L7: “Genetic differentiation and selection are stronger at broader geographic scales across Mascarene grey white-eye (*Zosterops borbonicus*; Milá et al., 2010) and barn swallow populations, (*Hirundo rustica*; Safran et al., 2016)” this sentence is not very clear to me.

P44. Figure S11: not quoted in the text

Appendix B

Genomic variation, population history and within-archipelago adaptation between island bird populations

We thank both reviewers for their extensive and insightful comments, and for the time taken to provide such detailed suggestions on this manuscript. Based on suggestions from Reviewer 1 we have provided additional references and context for our biological interpretations of linkage disequilibrium across the Berthelot's pipit populations, which is a commonly used analysis for understanding genetic diversity. To alleviate the concerns of both reviewers, we rerun our *TreeMix* models of population divergence across the 13 Berthelot's pipit populations, removing inference of a known rooting population and providing models for additional SNP window sizes. These results show patterns of divergence and admixture within and between archipelagos are consistent with our previous inferences. We have also removed the use of morphological PCA for the genotype-phenotype association analyses of our morphological candidate gene *ADAM12*, and simplified the methods and results in this section. As recommended by Reviewer 2, we have now fitted GLMM models for the *ADAM12* SNP genotypes to individual morphology measurements. We find head length to be weakly correlated with genotype at the *ADAM12* SNPs, which concurs with our previous conclusions from morphological PC4 (head and tarsus length relative to beak length). Overall our conclusions are robust to these additional analyses, and we believe that these changes, alongside the minor edits detailed below, substantially improve this manuscript. Below we provide a point-by-point response to each reviewer comment.

Reviewer 1

COMMENT: I think this is a very interesting paper on genetic variation within archipelago in the Berthelot's pipit, a passerine found in three archipelagos of the North Atlantic (the Canary islands, the Selvagens and the Madeiran archipelago). As far as I can tell, the analyses are well done, except for one major concern that I have regarding the analyses related to linkage disequilibrium (LD) (see major comment). I mostly have small comments and suggestions to make the manuscript a little bit clearer but this is a very polished version already and I'm recommending to accept this paper with minor revisions.

RESPONSE: We thank Reviewer 1 for their comments and are pleased that they found the study interesting and overall well conducted. We have edited the manuscript to address these concerns with particular refinement of the methods and conclusions regarding the population linkage disequilibrium analyses, and provide specific details on how we have addressed these comments below.

Major comments:

COMMENT: I was not convinced by the use of LD analyses to infer demography. This requires more explanations and references for the reader to understand how to interpret the LD - genetic distance plots in terms of demography, namely bottlenecks and founder events. Here are the sentences that would need more explanations and references:

RESPONSE: We have now provided additional explanation as requested. Patterns of linkage disequilibrium (LD) are commonly used to infer population size and bottleneck history in other species (e.g. humans, wolves and dogs) and we are careful now to cite this work throughout the manuscript to support our methods and conclusions. We also note that patterns of population level LD in the pipit are consistent with signatures found using other methods such as microsatellite

Genomic variation, population history and within-archipelago adaptation between island bird populations

markers and approximate Bayesian computation (Spurgin *et al.* 2014). We present these results here using RAD-seq markers to provide further insight into population size, bottleneck history and genetic diversity that cannot be revealed by the *TreeMix* analysis and to add population level detail to previous LD analyses conducted across broader spatial scales in this system (Armstrong *et al.* 2018).

We now provide additional clarity in the areas suggested by the reviewer:

P7.L45: “To further understand population demography, including bottlenecks and population size, LD was calculated for each population in Plink”

RESPONSE: Now updated to:

L305 – 316: “LD summarises both mutational and recombination history, whereby larger more outbred populations have rapid decay of LD between genetic markers compared to small inbred populations (Flint-Garcia *et al.* 2003). Patterns of LD have been used extensively to detect historic fluctuations in population size (N_e) and founder events in humans (Reich *et al.* 2001; Reich *et al.* 2009), selectively bred species such as Chinese Merino sheep, Xinjiang type (Liu *et al.* 2017) and wild species including European grey wolves, *Canis lupus*, (Pilot *et al.* 2014) and village dogs, *Canis lupus familiaris* (Shannon *et al.* 2015)..... To further understand population demography, including genetic diversity and population size in the Berthelot’s pipit, we calculate LD for each island population using Plink.”

P17.L34: “examine population level patterns of bottlenecks using linkage disequilibrium (LD) decay.”

RESPONSE: We agree that the bottleneck signature is not conclusive at the population level and have rephrased this sentence:

L670- 671: “examine population level patterns of genetic diversity using linkage disequilibrium (LD) decay.”

P17.L44: “Past colonisation history and associated bottlenecks are reflected in patterns of population level LD” please add references and explanations to help us understand the rest of the sentence “these indicate larger more outbred populations across the Canary Islands, but increasingly stronger signatures of inbreeding and reduced genetic diversity through bottlenecks and on small isolated islands”.

RESPONSE: L681 – 690 “..... ; sharp LD decay at proximate SNPs and low long range LD indicates larger more outbred populations across the Canary Islands, while shallow decay patterns and high long range LD indicate bottlenecks and/ or inbreeding and reduced genetic diversity (Pilot *et al.*, 2014; Shannon *et al.*, 2015). Our patterns of LD are consistent with reduced genetic diversity in the Madeira and Selvagens archipelagos and support previously reported patterns based on microsatellite and RAD data (see Spurgin *et al.* 2014; Armstrong *et al.*, 2018).

COMMENT: I was not convinced by your interpretations in terms of founder events and bottlenecks, in my opinion, this is not shown by your analyses.

RESPONSE: High levels of LD are the result of low genetic diversity. We have lower genetic diversity in the later colonised populations, which is particularly clear between the archipelagos and has been

Genomic variation, population history and within-archipelago adaptation between island bird populations

shown by previous analyses (see Spurgin *et al.* 2014; Armstrong *et al.* 2018). The population level patterns of LD support our previous inferences and we now make clearer reference to both our work and other studies that use these methods:

P17.L17: “Patterns of genetic diversity are consistent with signatures of founder events and geographic isolation.”

RESPONSE: We expand on this in the next paragraph as want to avoid external references in the first paragraph of the discussion.

P19.L30: “revealing that genetic diversity is largely shaped by colonisation events and related bottlenecks”

RESPONSE: We have now removed reference to bottlenecks in the conclusion.

COMMENT: Here is an additional comment on the LD plots:

P9.L34: Could you provide a quantification about the shallowness of the curve? What is the relationship between shallowness and demography (population size)?

I still think it is really interesting to link LD patterns with demography, and I am aware this has been done (see for instance the review by Waples *et al.* 2016: Waples, R.K., Larson, W.A., and Waples, R.S. (2016). Estimating contemporary effective population size in non-model species using linkage disequilibrium across thousands of loci. *Heredity* 117, 233–240.) but I was not convinced by the LD plots without any models or curve fitting.

RESPONSE: We did fit a local regression model which is represented by the curve in Figure 2B. Sorry that we did not highlight this clearly enough. This should now be clear in the methods as well as in the Figure 2 legend:

L317 – 318: “We fitted a local weighted linear regression (loess) curve to the r^2 data using R function ‘loess’, with 95% confidence intervals calculated.”

We did however notice that the shaded 95% confidence intervals and individual pairwise SNP r^2 values did not appear on the submitted manuscript version of the plot and this has now been rectified. Together these should aid interpretation of these plots. We decided not to quantify shallowness as we believe that visual inspection of these curves is best given the level of error we have shown. We have added clarification to the relationship between shallowness and population size as above L305 – 316, including:

L312-314: “The relationship between proximate SNPs reflects historic N_e and LD at distant SNPs reflects N_e in more recent times.”

Minor comments:

COMMENT: P5.L15: “To address these questions, we first use demographic analyses to examine colonisation history and gene flow across the species range” I am not sure that Treemix and the LD plots qualify as demographic analyses.

Genomic variation, population history and within-archipelago adaptation between island bird populations

RESPONSE: Changed to:

L153: “...we first use population genetic analyses to examine....”

COMMENT: P6.L4: How did you identify sex-linked loci?

RESPONSE: We used the same method to map Z chromosome as we did for the autosomes. RAD loci were mapped to the Zebra Finch genome (*Taeniopygia guttata*, version 3.2.4; Warren *et al.* 2010), with the assistance of the Berthelot’s pipit reference genome, as detailed in the methods to determine the genomic location of the SNPs.

COMMENT: P6.L5: Do you have missing data or did you do any filtering based on missing data?

RESPONSE: These SNPs were already trimmed for missing data and quality as detailed in Armstrong *et al.* 2018. We have added clarification on this in the methods:

L196 - 203: “The initial ddRAD library was generated using the protocol by DaCosta and Sorenson (2014) which assigns RAD reads to samples based on an 8 bp barcode sequence and retains the read with the highest quality score. Loci that could not be confidently genotyped in more than four samples and those where 10% or more calls were missing or ambiguous were treated as missing data in the “Berthelot’s” library. The “All Pipits” dataset containing all Berthelot’s pipit and tawny pipit samples was filtered to contain SNPs from RAD loci that were successfully genotyped in 100% of individuals, removing loci that contained SNPs with >2 alleles.”

COMMENT: P6.L60: supplementary methods – typo last sentence “by was dominated”

RESPONSE: Changed to “but was dominated by”.

COMMENT: P7.L7: Why do you think there could be gene flow between islands? This would help to justify the use of Treemix.

RESPONSE: We have previously assessed admixture across the 13 island populations of the pipit and see an east-west population structuring across the Canary Islands, alongside low levels of within-archipelago divergence (see Armstrong *et al.* 2018). We therefore think that it is possible that gene flow between closely located islands may be responsible in part for these patterns. We have added clarification of this point:

L240 – 248: “Strong population structuring exists between archipelagos of the Berthelot’s pipit supporting our previous inference of absence of contemporary gene flow at broad scales in this system (Armstrong *et al.* 2018). We have reported weak east-west population structuring between populations within the Canary Islands (Armstrong *et al.* 2018), but it is unknown whether this is due to contemporary gene flow between closely located islands or weak population divergence since colonisation. We implement *TreeMix* at these different population scales with the aim of further understanding the evolutionary processes behind the patterns of population structuring we see.”

Genomic variation, population history and within-archipelago adaptation between island bird populations

COMMENT: P7.L20 “within population”: it would help to remind what your populations are, i.e. one population per island except in Tenerife with two populations in the lowlands and in the highlands.

RESPONSE: Added

L262 – 264: “Using *TreeMix* v 1.13 (Pickrell and Pritchard 2012), we inferred a tree in which populations (i.e. one population per island except in Tenerife with two populations, one in the lowlands and one in the highlands) may maintain gene flow after they split from a common ancestor.”

COMMENT: P7.L22 “rejecting migration events when the following event did not significantly increase the likelihood of the model”: what number would you consider to be a significant increase of the likelihood?

RESPONSE: Sorry this was a poor choice of wording. We have now changed this:

L269 – 271: “We modelled several scenarios allowing zero to eight migration events, discounting migration events when the relative increase in model likelihood was <1%.”

We have also altered the Figure S5 to show the total likelihood values to allow easier comparison of relative likelihood scores.

COMMENT: P7.L25: I was not convinced by the choice of the number of SNPs in one block, I find the number 45 to be quite specific. Also, the results with a block of 50 SNPs (and 50 seems to be very close to 45) are quite different for the Canary populations i.e. no east-west trend as discussed in the results and discussion.

RESPONSE: The window sizes (20-50 SNPs) were chosen to reflect the range (10-20 Mb) considered by the authors of the *TreeMix* package (Pickrell and Prichard 2012). The residual fit, used to determine the fit for each pair of populations, was poorer for many populations using 50 SNP windows compared to the tree produced using 45 SNP windows. However, we can see your point that 45 and 50 are extremely close and now instead present 20, 50 and 80 SNP windows and report specific results for 50 SNP windows. To correspond with this change we now report the F3 results for windows of 50 SNPs to match the new *TreeMix* plot.

Broad population demographic patterns are similar across these window sizes. In all three window sizes the furthest eastern islands and furthest western islands are together on the same branch with some variation in divisions of branching for the more central islands. We do highlight that there is some variation in these details on L433 - 435: “Tree topography was broadly robust to window size, but there were minor differences within the Canary Islands including the source populations for the Madeiran and Selvagens archipelagos (Fig. S3).”

COMMENT: P7.L39: typo “Berthelot’s”

RESPONSE: changed to “Berthelot’s pipit”.

Genomic variation, population history and within-archipelago adaptation between island bird populations

COMMENT: P7.L44: “PCA=within archipelago; LD=within populations”. Please clarify this sentence.

RESPONSE: We have expanded this to say:

L303 – 305: “PCA was filtered according to MAF (SNPs with MAF < 0.03 excluded) within archipelagos, LD analysis was MAF-filtered (SNPs with MAF < 0.03 excluded) within populations”

COMMENT: P8.L4: What is the general principle of EigenGWAS? What is the measure calculated and how does this relate to selection? What is the link with principal components mentioned in the results?

RESPONSE: We agree that additional details are needed in this section, and have added the following to the manuscript:

L332 – 347: “EigenGWAS performs a PCA to generate gradients of population structure, then assesses each genetic marker individually along these axes of population variation. EigenGWAS provides genomic inflation factor corrected P values (λ_{GC}) - with the significance threshold determined by Bonferroni-correction - to control for genome-wide population stratification and drift. Loci above this threshold are putatively under selection across the gradient of population structure (see PCAs, Figure 3).”

COMMENT: P8.L19: Please indicate the program that you used for the morphological PCA.

RESPONSE: In response to concerns of Reviewer 2 this analysis has now been removed, and the methods simplified. See detailed comments below.

COMMENT: P8.L33: 9,960 SNPs without any filtration seems to be very low. Did you filter by missing data?

RESPONSE: See comment above on data filtering.

COMMENT: P8.L55: I would have liked to see the results of the PCA first before Treemix analyses as it is an exploratory analysis without any model and assumptions.

RESPONSE: Within this paper the PCAs are predominantly used as part of the EigenGWAS analyses for loci under selection. We looked at restructuring but felt that presenting PCA results within archipelagos prior to considering colonisation and gene flow at broad scales would further confuse the reader. We have clarified the link between the PCA and EigenGWAS so that this ordering has greater clarity.

COMMENT: P9.L12: “long branch length”, I don’t think this branch is that long compared to the very long branches of the Selvagens and Madeiran archipelagos.

Genomic variation, population history and within-archipelago adaptation between island bird populations

RESPONSE: Good point, changed to “Generally weak drift is observed across the Canary Islands, with the longest within archipelago branch lengths in El Hierro and La Graciosa.”

COMMENT: P9.L30: “F3 results suggest weak admixture events subsequent to branch divergence may exist between Madeira and Porto Santo and Fuerteventura/Lanzarote and La Graciosa.” Maybe you could add that this is consistent with the geographic distances between the islands. It’s reassuring that you did not infer some possibility of admixture between very distant islands.

RESPONSE: Indeed, it is reassuring that both the F3 admixture results and lack of support for gene flow in the *TreeMix* models point to absence of migration between distant populations. We have added the following:

L445- 447: “This is consistent with geographic distance between islands, suggesting no admixture between geographically distant populations.”

COMMENT: P9.L41: How about presenting a PCA with all three archipelagos together ?

RESPONSE: PCA analysis across the Berthelot’s pipit range was previously analysed by Armstrong *et al.* (2018) and we make reference to these results on L135-136: “Analysis showed strong genetic structure among, but not within, archipelagos...”. These conclusions were based on PCA and ADMIXTURE results at broad spatial scales in this system.

COMMENT: P9.L45: I would rather say “PCA of the Madeiran archipelago separated Deserta Grande from Madeira and Porto Santo along the first axis, and separated Porto Santo from Deserta Grande on the second axis, with a weak gradient from Madeira to Deserta Grande to Porto Santo (Fig. 3B).”

RESPONSE: Changed.

COMMENT: P9.L57: What is the link between EigenGWAS and the PCs? Please explain this in the methods as well to make the results more understandable.

RESPONSE: See comment above on principles of EigenGWAS. We have added detail to the methods regarding the link between PCAs and EigenGWAS outlier detection.

COMMENT: P10.L24: How close to the SNP 219s24 are the genes *WDH1* and *GCH1*? Please indicate the chromosome here as well.

RESPONSE: Done – we have added details on SNP location relative to these genes in the results section:

L507- 510: “The two closest genes to the Canary Island SNP, 219s24, are *WDHD1* and *GCH1*, which are involved in DNA binding/repair and enzyme synthesis, respectively (Table 1). This SNP maps to chromosome 5, with *WDHD1* 2,071 bp upstream and *GCH1* 6,252 downstream.”

Genomic variation, population history and within-archipelago adaptation between island bird populations

COMMENT: P12. Figure2A: I would suggest to add colours or bars with colours for archipelagos grouping the three islands of Madeira and with eastern and western and Tenerife groups of the Canaries so that we can more easily read the figure and relate it with the text.

RESPONSE: Good idea, we have coloured the population labels to match the Figure 2B LD plots as below:

COMMENT: Figure2B: I don't see the link between LD and Treemix analyses so I would not put them in the same figure. Careful, the "shaded band indicating 95% confidence intervals" does not appear on the plots.

RESPONSE: Our aim was to link the results of population history together (LD – population size and bottlenecks, *Treemix* – colonisation, gene flow and drift) and then keep the selection-related plots separate. We think this works, but that it is ultimately an editorial decision and are happy to change these to separate plots if preferred in light of this comment. Thank you for highlighting the missing 95% confidence intervals, this has been rectified in the new submission.

COMMENT: P15. Figure 4: The Z chromosome is not present here? If not, could you indicate why? What about the chromosomes 1A and 4A? When reading the scale ($-\log_{10}(p)$), it is not clear what quantity is calculated, and this should be explained in the methods (see previous comment on EigenGWAS).

RESPONSE: There are markers mapped to all large chromosomes including the Z chromosome which are represented by the points in Figure 4A and B. We have added additional labels for these large chromosomes in the updated plot. Additional information has been provided in the methods on how the EigenGWAS *P* value is calculated – see major comment from Reviewer 2.

COMMENT: P16.I39: Typos "Because ADAM12 has been shown to play a role" and "is therefore a potential candidate"

Genomic variation, population history and within-archipelago adaptation between island bird populations

RESPONSE: Changed.

COMMENT: P16.I42: Typo “A principal component analysis”

RESPONSE: Changed.

COMMENT: P16.I43: Please show the PCA variable correlation plots to support the relationships between the PCs and the variables mentioned in the text and show the morphological PCA in the supplementary materials.

RESPONSE: Analysis simplified in response to Reviewer 2 suggestions, and to avoid many additional supplementary figures. See specific changes below in Reviewer 2 major comments.

COMMENT: P16.I48: Are the results the same for the other SNP?

RESPONSE: See response to Reviewer 2 – we agree that adding both sets of results is a more transparent way to present these results and have provided these for the new GLMM that we present.

COMMENT: P17.I22: I may have missed something but it seems to me that the sentences “no overlap between candidate SNPs identified from previous analyses at a broader spatial scale i.e. between archipelagos” and p18.L25 “It is worth pointing out that our outlier SNP in the Canary Islands, 219s24, was in the same broad genomic region on chromosome 5 as a set of SNPs previously identified as being under selection and associated with bill length across the range of Berthelot’s pipit (Armstrong et al., 2018)” are conflicting.

RESPONSE: We have clarified the distinction between these results by adding greater detail to the second quote:

L734- 736: “It is worth pointing out that while we do not detect selection for the same SNPs within archipelagos as between archipelagos, our outlier SNP in the Canary Islands, 219s24, was in the same broad genomic region on chromosome 5 as a set of”

COMMENT: P17.L37: “Tree topography suggests initial divergence of Berthelot’s pipit from the tawny pipit may have been to the central or eastern Canary Islands (Fig. S2A)”. I don’t think we can say that from figure S2A as the branches of the Canary form a polytomy. I don’t think you can say that from figure 2A as well as from what I understood it is not rooted.

RESPONSE: This is a fair point, this result is tenuous and we have now removed the assumption of tree rooting for the Berthelot’s pipit population tree. We have removed references to specific regions of the Canary Islands for initial divergence/ colonisation and instead just make reference to initial colonisation of the Canary Islands from Figure S2A. Patterns of population divergence are consistent when using an unrooted tree across the Berthelot’s pipit populations and so our conclusions regarding divergence across the Berthelot’s pipit are independent of this inference.

Genomic variation, population history and within-archipelago adaptation between island bird populations

COMMENT: P18.L8: Please check all the “within” to decide with the italic style.

RESPONSE: We agree that this was confusing and have now removed italics throughout.

COMMENT: P18.L29: “Further research, with a higher density of SNPs, is needed to identify the importance of this genomic region for adaptation in the pipit system.” Yes I agree! I think you have a relatively low number of SNPs to perform genomic scans compared to what can be done now from whole genome sequences, so it’s important to mention that. You may also elaborate on the very few number of outlier loci that you detected. Do you think you could have missed important outlier loci compared to if you had used whole genome sequences?

RESPONSE: We agree, and work is currently being undertaken using whole genomes to follow up on these regions and to identify further loci under selection at broad scales and between recently separated populations.

L740 – 745: “Using our marker set, we detected only four loci under selection in our analyses. We see low levels of genetic variation across Berthelot’s pipit populations as a result of colonisation history and bottlenecks but it is also likely that we have no marker coverage in other regions of the genome that are under selection. Therefore, future studies are needed to uncover greater detail on the loci and hence traits that are under selection in this system.”

COMMENT: P19.L7: “Genetic differentiation and selection are stronger at broader geographic scales across Mascarene grey white-eye (*Zosterops borbonicus*; Milá et al., 2010) and barn swallow populations, (*Hirundo rustica*; Safran et al., 2016)” this sentence is not very clear to me.

RESPONSE: Changed to:

“The strength of genetic differentiation and selection increases with geographic distance between Mascarene grey white-eye (*Zosterops borbonicus*; Gabrielli et al. 2020) and barn swallow populations, (*Hirundo rustica*; Safran et al., 2016) and between lake and ocean populations in brown trout (*Salmo trutta*; Meier et al., 2011) and Atlantic salmon (*Salmo salar*; Vincent et al., 2013).”

COMMENT: P44.Figure S11: not quoted in the text

RESPONSE: Reference to this figure is now made in the section discussing the genotype-phenotype association of *ADAM12* SNPs.

Reviewer 2

COMMENT: In your paper “Genomic variation, population history and within-archipelago adaptation between island bird populations” you perform population genetic analyses and search for loci under divergent selection in Berthelot’s pipits sampled across three Macaronesia archipelagos. In general, I enjoyed reading your manuscript although it was at times hard to follow given the many analyses performed and the large supplement. I have a few major points that you may want to consider (order is arbitrary and not by importance):

RESPONSE: We thank Reviewer 2 for their thoughtful and detailed comments on this manuscript. We are glad that the reviewer found the manuscript interesting and have tried to improve the clarity of analysis results as you have mentioned in your comments so that the conclusions may be better understood by the reader. We respond to each comment in turn below.

COMMENT: 1. Please provide some more details on the RAD-data, e.g. coverage per individual and software used. I think that these minimal details should be provided here rather than referring to Armstrong et al. (2018).

RESPONSE: Please see response to Reviewer 1 – we have included specific details on the bioinformatics performed to generate the initial RAD libraries used in this study.

COMMENT: 2. You mention reduced genetic diversity on the smaller islands several times in the manuscript but never estimated π . This should be possible with your RAD-data and would be a nice addition.

RESPONSE: Nucleotide diversity was calculated per SNP and per locus for each of the archipelagos, and separately for Deserta Grande island, in Armstrong *et al.* 2018. In the interest of keeping the manuscript succinct we avoided replicating these analyses here with just a few additions at population level. We now make more explicit reference to these previous results to back up our inferences of genetic diversity from the LD analysis. See comments to Reviewer 1 for details.

COMMENT: 3. EigenGWAS: I never used that program but as far as I understand it takes the PC1 scores of each individual as the dependent variable and each SNP (used for the PCA) as the independent variable. I guess that it does not correct for relatedness.? Thus, it should be similar to the method described in Duforet-Frebourg et al. (2016) or used in Knief et al. (2019). It is basically a representation of the loadings, right? Thus, it (A) is not too surprising that the SNPs you detect are also those with the highest F_{ST} (Fig. 5). (B) I was wondering if in the Madeiran archipelago the two significant SNPs generate PC1 because they are in high LD ($r^2 > 0.982$). Would it not be more informative to use the LD-pruned SNP set here? Also, I was wondering whether a Bonferroni-corrected significant threshold is conservative here given that SNPs are in LD. The method implemented in Gao et al. (2008) may be useful here. Last, throughout the manuscript you write “SNPs under selection” (except for the Conclusions paragraph). I would rather say “SNPs putatively under selection” because EigenGWAS does not proof selection.

Genomic variation, population history and within-archipelago adaptation between island bird populations

RESPONSE: Yes this is correct. We have added additional detail on the theory of EigenGWAS in the methods so this should be easier to follow (see response to Reviewer 1). We remove closely related individuals for both the PCA and EigenGWAS analysis, so that the PCs are not influenced by uneven sampling of individuals across populations. EigenGWAS implements a genomic inflation correction for P values (λ_{GC}) to correct for population stratification (i.e., drift), to avoid interpreting ancestry-informative markers as loci under selection. Then, using these corrected P values we test for SNPs putatively under selection along that PC axis. The Bonferroni corrected significance threshold is conservative, but we believe this is the best approach for exploratory analyses such as these. It is useful (and standard) to view SNPs in context of each other which would not be possible with an LD-pruned set.

We agree that it is not surprising that the EigenGWAS results are highly correlated with F_{ST} , as they are both reliant on allele frequencies. We present the F_{ST} results for two reasons. First, many other studies rely solely on F_{ST} to detect such outliers and so we feel it is important to show that both analyses reveal the same outliers. Secondly, it is an interesting visual representation of the fact that very different SNPs are under divergence within the Madeiran archipelago compared to the Canary Islands which is not presented by the EigenGWAS plots.

Finally, we agree that we cannot be confident that these SNPs are under selection and have changed our wording to “SNPs putatively under selection” throughout.

COMMENT: 4. Genotype-phenotype associations: Why do you not fit your relatedness matrix as a random effect (and use all individuals, including the related ones)? Accounting for population is not enough to control the type I error rate here. You used lme4 for fitting the LMMs (not described in the Methods section though and only visible in the provided code) but I think that with the R-package lme4qtl you should be able to do this (Ziyatdinov et al. 2018; although I have never used that specific package, I know that lme4 can be modified to handle any relatedness matrix: this function was called relmatmm and can be found via Google).

I am wondering why you use principal components and not the raw phenotypes in your genotype-phenotype associations. The PCs are hard to interpret (except for PC1, which mostly reflects size) and I would thus suggest that you use the raw phenotypes as dependent variables and – in case you want this – correct for size by fitting PC1 as an independent variable in your model. Also, using all these PCs may look like data dredging.

RESPONSE: Given the low relatedness values in our filtered dataset, we believe that fitting the relatedness matrix as a random effect is unnecessary and would result in overfitting given our sample size. Further, we have been very careful throughout to emphasise the tentative nature of our results in relation to morphology.

However, we do agree on the PCA, and we have now simplified these analyses to remove the morphology principal components and instead investigate effect of *ADAM12* genotype on three morphological measurements of head length, wing length and bill length. We find significant effect of genotype on head length and report results for these three GLMMs.

COMMENT: 5. Table S1: Although the supplement is already extensive, I think that you should split that table into two. Table S1A: The part above the dashed line (i.e. which trimming/filtering steps used for which analyses). Table S1B: Put the different populations as columns and provide the total numbers of individuals and SNPs as rows. Add rows for each filtering step and give the number of individuals/SNPs removed (or kept). Consider adding these two tables to the main text.

Genomic variation, population history and within-archipelago adaptation between island bird populations

RESPONSE: We have separated this table as recommended but were unable to find a practical way of adding population level information to the table. Of our analyses, most have been conducted either using the Berthelot's full data set (*TreeMix*, MAF) or with the archipelago level datasets (PCA, EigenGWAS, Loci F_{ST}) as described in the methods and therefore full information is provided already in this table. It is complex to provide full details for Berthelot's pipit pairwise F_{ST} as we tested all 78 potential combinations of populations and hence we provide a range for the SNP numbers in the table. We could provide full details for LD and Relatedness results by population but as the SNP ranges are small we feel it confuses the table without separating into a further supplement.

We have kept these tables in the supplementary results to keep the manuscript as concise as possible but can move if the reviewers/editors feel strongly about this.

COMMENT: 6. LD-analysis: Please provide details on the local regression model used for fitting the regression lines in the Methods section. In Fig. 2 I was wondering why there is always a "dip" in the regression line between 25 and 50 kb. Does the local regression model provide estimates such that you can statistically show that "baseline" LD is higher in populations on smaller islands (some kind of intercept)? Why do you not show all populations in Fig. 2 (rather than having an additional Fig. S7)? I think that adding the Selvagens to Fig. 2 is necessary.

RESPONSE: We have now added details regarding the fitted regression line in the methods section (see response to Reviewer 1). We were also intrigued by the consistent dip seen in the regression line between 25 and 50 KB and are not sure why we see these patterns for many populations. We have added this to the discussion:

L687- 690: "One common feature of population level LD decay is a dip in the regression between 25 and 50 KB followed by a rise. This pattern has been found in previous studies of LD in this system but we are unable to determine a biological explanation for this."

Presenting the LD plots for all populations in the main paper takes up a lot of room for not much additional information as most populations in the central Canary Islands are very similar and there was no presentable way to put these all on one plot due to substantial overlap of lines. We have added SG to Figure 2.

COMMENT: 7. *TreeMix*: I am not very familiar with this software but to me the branches in Fig. 2A and even more so in Fig. S2A appear very short. However, you base some conclusions (i.e. that TF was colonized first) partly on the branch lengths. Is this justified?

RESPONSE: Indeed, branch lengths between populations within the same archipelago are short compared to those between archipelagos. We agree that caution must be taken when making conclusions from these branch lengths and now do so. See response to Reviewer 1, above.

COMMENT: 8. Provide line numbers for the next revision. My minor comments are just "approximate" line numbers.

I hope you find these comments helpful and best wishes.

Genomic variation, population history and within-archipelago adaptation between island bird populations

RESPONSE: We apologise, this was an error and additional continuous line numbers have been added throughout the manuscript revision in addition to the page-by-page line numbers.

Minor comments:

COMMENT: P3L5: “evolutionary processes”

RESPONSE: Changed.

COMMENT: P3L13 and throughout: Could you reformulate “very limited evidence”? Do you have little evidence (i.e. there may or may not be gene flow) or little (if any) gene flow?

RESPONSE: We now make clear that we are referring to limited evidence that gene flow is occurring.

COMMENT: P3L32: Why not use “depends”?

RESPONSE: Changed.

COMMENT: P3L38: “Studies on humans”

RESPONSE: Changed.

COMMENT: P3L39: “important for our understanding”

RESPONSE: Changed.

COMMENT: P3L41: “insights”

RESPONSE: Changed.

COMMENT: P4L13: please reformulate “evolutionary significant adaptations”. Significant is a word used in statistics. Are there also “evolutionary non-significant adaptations? Likewise “evolutionary important adaptations” (L21) sounds strange.

RESPONSE: Good point, we have reframed this throughout to “adaptive evolution”.

COMMENT: P4L14: replace “subsequent” with “other”

RESPONSE: Changed.

Genomic variation, population history and within-archipelago adaptation between island bird populations

COMMENT: P4L57: So it is a question of parallelism at the molecular level? Do birds from different archipelagos from similar habitats resemble each other?

RESPONSE: This is an important question, and one we are interested in. However quantifying habitat similarity is a large amount of work, and something for future study.

COMMENT: P5L13: You do not quantify selection (differentials). So you cannot address the question of “how strong”.

RESPONSE: This is a fair point, and we have reduced the emphasis of quantifying the strength of selection and instead highlight the comparison between detectability at between archipelago and within-archipelago levels. We now ask: “Can we detect signatures of selection across recently diverged populations within archipelagos?”

COMMENT: P5L14: You have only a handful of (morphological) traits.

RESPONSE: We are only interested in body size measurements to follow up on the *ADAM12* loci and for this we feel we can get an idea of what morphological features are potentially linked to this loci. Indeed we are not able to follow up any further on the other loci under selection as they are not linked to measurable traits.

COMMENT: P5L46: Remove “also”

RESPONSE: Changed.

COMMENT: P5L54: I do not understand what is meant by “fewer than 4 ambiguous genotypes”.

RESPONSE: There are several reasons why a genotype is flagged as ambiguous. This is discussed in the DaCosta and Sorensen paper which we now cite here for clarity and have now reworded this section to provide additional filtering information.

COMMENT: P5L58: “start (s)”

RESPONSE: The s actually occurs in the SNP name as you can see in Figure 4.

COMMENT: P6L14: “included in the”. You should maybe write here how many were removed.

RESPONSE: Changed typo. Specific details on numbers of individuals removed are now given in the results.

COMMENT: P7L12: How sensitive are your analyses to these thresholds?

Genomic variation, population history and within-archipelago adaptation between island bird populations

RESPONSE: We did not specifically test different thresholds for standard trimming steps but instead used these to reduce the influence of uneven sampling (both genetically and in the population) on these analyses based on pre-decided thresholds which are also similar to those used by Armstrong *et al.*, (2018) to which we make substantial comparison of results.

COMMENT: P7L22: Refer to Tab. S1 here.

RESPONSE: Done.

COMMENT: P7L50 and throughout: Check spelling of “principal component analysis”

RESPONSE: Changed from “principle” to “principal” throughout.

COMMENT: P7L53: “close relatives and SNPs with a MAF < 0.03 were removed”

RESPONSE: Changed.

COMMENT: P7L54: “filter based on LD”

RESPONSE: Changed.

COMMENT: P7L60: Maybe briefly describe what EigenGWAS does.

RESPONSE: See response to Reviewer 1, additional background detail provided.

COMMENT: P8L21: Which software did you use for fitting LMMs?

RESPONSE: We used lme4 and have added this detail to the methods.

COMMENT: P8L25: Replace “run” with “fitted”

RESPONSE: Done.

COMMENT: Fig. 2 and throughout: You could color the populations stemming from the same archipelago and use different symbols but the same color for populations within archipelagos.

RESPONSE: See response to Reviewer 1 – we have changed the labels in Figure 2A to match the LD plots.

COMMENT: P9L30-32: I do not see this in Tab. S2. Also, these are a lot of tests.

Genomic variation, population history and within-archipelago adaptation between island bird populations

RESPONSE: We have explained how to interpret this result in more detail here and have reduced the number of tests reported here to remove marginal results. All combinations of populations have an F_3 statistic, of these we now report tests that had a negative Z-score < -0.5 , and therefore may be of interest for admixture.

COMMENT: P9L36: “Thus, across all populations”

RESPONSE: Done.

COMMENT: P9L38: How do you define “baseline”

RESPONSE: We have now added the range of interest in the results L450: “here considered to be 100-150 Kb”.

COMMENT: P10L28: Are these SNPs located in the coding region? Synonymous?

RESPONSE: These SNPs map to intronic regions of the *ADAM12* gene, which we now clarify on line 512.

COMMENT: P13L4: I cannot see the shaded bands

RESPONSE: This has now been rectified in the new figure, sorry.

COMMENT: P15L48: “ $P < 1.70 \times 10^{-5}$ ”

RESPONSE: Changed, thank you for spotting this.

COMMENT: Fig. 5: Think about moving it to the supplement

RESPONSE: We have kept this figure as it visually shows a) that the EigenGWAS SNPs are the same as those in high F_{ST} (which maybe isn't a surprise since they both rely on allele frequencies) and b) that there are very different groups of SNPs under divergence within the two archipelagos.

COMMENT: P16L30: Where does the number “3531” come from? Why not all 4470 SNPs?

RESPONSE: This just includes the mapped SNPs so that a direct comparison can be made with the EigenGWAS results, and this is stated in the methods table S1A “* = loci successfully mapped to chromosomes with $MAF > 0$.” and we have added detail to the Figure 4 legend.

COMMENT: P16L39: “ADAM12 has been” ... “is therefore a potential”

Genomic variation, population history and within-archipelago adaptation between island bird populations

RESPONSE: Changed.

COMMENT: P16L49: Based on what did you decide showing results from SNP1585s112?

RESPONSE: The aim was to simplify the presentation of the results as they were almost identical. Although the results are very similar, we agree that adding both sets of statistics in the results is more transparent. These results are now presented instead for the individual morphology variables.

COMMENT: P16L52+55: I think it is “Fig. 11”

RESPONSE: Changed.

COMMENT: P16L52: “Homozygous”

RESPONSE: Changed.

COMMENT: P17L39: From the Canary Islands, not from the mainland

RESPONSE: Berthelot’s pipit is endemic to Macaronesia and we have no evidence of contemporary or historic populations of the Berthelot’s pipit populations on the mainland. We think it could be more confusing to add reference to the mainland.

COMMENT: P17L45: “larger and more”

RESPONSE: Changed.

COMMENT: P17L46: “islands, and increasingly”

RESPONSE: Changed.

COMMENT: P17L47: remove “through bottlenecks and” (not shown)

RESPONSE: Done.

COMMENT: P17L56: is it “strong” or “high” LD. Replace “of” with “and”

RESPONSE: Here we are trying to link the patterns of LD with inferences of genetic diversity. The genetic signature is strong and it suggests reduced genetic diversity. We have reworded this from “presence of strong LD signatures of reduced genetic diversity following bottlenecks in the Madeiran archipelago and Selvagens are consistent with absence of contemporary gene flow between archipelagos.”

Genomic variation, population history and within-archipelago adaptation between island bird populations

To: “shallow LD decay and reduced genetic diversity in the Madeiran archipelago and Selvagens are consistent with absence of contemporary gene flow between archipelagos.”

COMMENT: P18L1-7: Maybe remove?

“Given that levels of divergence differ between pairs of closely located Berthelot’s pipit populations, we cannot discount the possibility of weak gene flow slowing the accumulation of genetic divergence between some pairs of recently separated populations which we have been unable to detect using our marker set “

RESPONSE: We feel that discussion of these results should include the limitation of our marker set to detect signatures of weak gene flow between islands. Hopefully we will be able to be more confident on this in our future work with increased markers, but we feel this caveat is an important consideration for the reader here.

COMMENT: P18L37: “that the genotype”

RESPONSE: Changed.

COMMENT: References: standardize lower and upper cases

RESPONSE: Done.

COMMENT: P27L15: Why did you use windows of 100 SNPs and not some distance measure?

RESPONSE: The purpose of this step is to account for LD in the analysis. The *TreeMix* package uses the command -k to determine the number of SNPs in a sampling block, which can be translated to an average distance measure but *TreeMix* doesn’t implement a distance option for this step. Averaging over windows is commonly used (see for example Foote and Morin 2016; Pedersen *et al.* 2018).

COMMENT: P27L16: “but was dominated”

RESPONSE: Changed.

COMMENT: Tab. S2: Not ordered by Z-scores. Could you make this more readable, e.g. by ordering the triplets alphabetically? In these analyses, do you need to account for multiple testing?

RESPONSE: We have clarified how this table is ordered more clearly “Results are ordered by Z-score within population groups.” Admixture events with a negative Z-score are reported by population and then ordered by population Admixture Z-scores (the most significant first). We think that this makes population comparisons easiest.

COMMENT: Tab. S4: Provide variance explained by the PCs. “Principal component”

Genomic variation, population history and within-archipelago adaptation between island bird populations

RESPONSE: This is no longer relevant now that the analysis has been simplified to include individual morphological variables.

COMMENT: Fig. S2B: Could you explain the high values between TAW, M, PS and DG?

RESPONSE: The *TreeMix* models across the three archipelagos poorly fitted populations in the Madeiran archipelago (as can be seen by the residual plots), and we are unsure of the reason for this. We were unable to improve the residual fit for these three populations by adding migration events. We have now highlighted this in the results L408-411:

“Populations in the Madeiran archipelago had poor residual fit with the Tawny pipit (Fig. S2), suggesting that these populations may be more closely related than is presented by the best fit tree. However, adding migration events did not improve the residual fit of these models.”

References referred to in response document:

- Armstrong, C., Richardson, D.S., Hipperson, H., Horsburgh, G.J., Küpper, C., Percival-Alwyn, L., Clark, M., *et al.* (2018). Genomic associations with bill length and disease reveal drift and selection across island bird populations. *Evolution Letters* **2**:22–36.
- DaCosta, J.M. and Sorenson, M.D. (2014). Amplification biases and consistent recovery of loci in a double-digest RAD-seq protocol. *PLoS ONE* **9**.
- Flint-Garcia, S.A., Thornsberry, J.M. and Edward IV, S.B. (2003). Structure of Linkage Disequilibrium in Plants. *Annual Review of Plant Biology* **54**:357–374.
- Foote, A.D. and Morin, P.A. (2016). Genome-wide SNP data suggest complex ancestry of sympatric North Pacific killer whale ecotypes. *Heredity* **117**:316–325.
- Gabrielli, M., Nabholz, B., Leroy, T., Milá, B. and Thébaud, C. (2020). Within-island diversification in a passerine bird. *Proceedings of the Royal Society B: Biological Sciences* **287**.
- Liu, S., He, S., Chen, L., Li, W., Di, J. and Liu, M. (2017). Estimates of linkage disequilibrium and effective population sizes in Chinese Merino (Xinjiang type) sheep by genome-wide SNPs. *Genes & Genomics* **39**:733–745.
- Pedersen, C.E.T., Albrechtsen, A., Etter, P.D., Johnson, E.A., Orlando, L., Chikhi, L., Siegismund, H.R., *et al.* (2018). A southern African origin and cryptic structure in the highly mobile plains zebra. *Nature Ecology and Evolution* **2**:491–498.
- Pickrell, J.K. and Pritchard, J.K. (2012). Inference of population splits and mixtures from genome-wide allele frequency data. *PLoS Genetics* **8**.
- Pilot, M., Greco, C., Vonholdt, B.M., Jędrzejewska, B., Randi, E., Jędrzejewski, W., Sidorovich, V.E., *et al.* (2014). Genome-wide signatures of population bottlenecks and diversifying selection in European wolves. *Heredity* **112**:428–442.
- Reich, D., Thangaraj, K., Patterson, N., Price, A.L. and Singh, L. (2009). Reconstructing Indian population history. *Nature* **461**:489–494.

Genomic variation, population history and within-archipelago adaptation between island bird populations

Reich, D.E., Cargili, M., Boik, S., Ireland, J., Sabeti, P.C., Richter, D.J., Lavery, T., *et al.* (2001). Linkage disequilibrium in the human genome. *Nature* **411**:199–204.

Shannon, L.M., Boyko, R.H., Castelhano, M., Corey, E., Hayward, J.J., McLean, C., White, M.E., *et al.* (2015). Genetic structure in village dogs reveals a Central Asian domestication origin. *Proceedings of the National Academy of Sciences of the United States of America* **112**:13639–13644.

Spurgin, L.G., Illera, J.C., Jorgensen, T.H., Dawson, D.A. and Richardson, D.S. (2014). Genetic and phenotypic divergence in an island bird: Isolation by distance, by colonization or by adaptation? *Molecular Ecology* **23**:1028–1039.

Appendix C

I thank the authors for taking into account my suggestions as well as the ones from reviewer 2. I think that the manuscript has been subsequently improved and I am very happy with the answers you provided. However, in my opinion, one concern remains regarding the Linkage Disequilibrium analyses. I find them a lot clearer now that you have provided context and I have carefully read the references that you added which I think are extremely interesting. I agree with you that the Selvagens have a higher baseline LD, as the Madeiran island populations, that could suggest a reduced genetic diversity and small population size. My question concerns what you mean by “shallow” and “sharp”. When looking at Figure 2B, I see that the curves of the Madeiran islands (green) and the Selvagens (orange) are sharper than the curves of the Canary Islands (purple), since the initial slope is more negative in the curves of the Madeiran and Selvagen islands than in the curves of the Canary Islands. Yet, you say that “LD decay was shallow across the three Madeiran islands”, so I did not understand well what you mean by sharp and shallow. In addition, still concerning the LD analyses, in all the papers that you cited, simulations were conducted to confirm that the shape of the LD curve was consistent with a bottleneck. I think this would be a very interested venue, but as you already did many analyses and the paper is close to publication, maybe you could only mention that a simulation-based approach would help confirming that these LD plots can be inferred in terms of bottlenecks. Finally, another interesting analysis would be to use your r^2 estimations to infer population size and variation in population size over time from Sved Equation (1971) as in Liu et al. 2017 or Pilot et al. 2014 if you have enough data for this. I’m particularly interested in knowing if you can decipher a long term low population size because the islands are small from a recent bottleneck due to founder event as in Pilot et al. 2014.

Here are some additional minor changes that I’m suggesting. Please note that the line numbers correspond to the manuscript with track changes:

L 51. change is dependent to depends upon

You removed ‘demography’ in the conclusion, so should you remove it also here?: L 271. 289.

L 290. we calculated

L 374. the Tawny pipit rooted tree obtained without adding migration events

L 378. including just → limited to

Figure 1: maybe add the three colours of the different archipelagos, for example as a frame in each of the archipelago name? And in that case don’t use the green in the zooming frame as it is already used for the Maldeiran archipelago.

Figure 2: add the Selvagens in the figure caption.

Figures S9 and S10: indicate the chromosome Z

742. whose

I want to congratulate again the authors for this manuscript that I really much enjoyed reading and I hope that they will find my questions about linkage disequilibrium analyses relevant.

Appendix D

Genomic variation, population history and within-archipelago adaptation between island bird populations

Associate Editor

COMMENT: The reviewers are much happier with this iteration of your work and we are grateful that you've carefully engaged with their concerns. That said, there are a number of matters that need addressing before your paper may be considered ready for acceptance. Firstly, there are a couple of relatively major issues you'll need to tackle, along with a range of smaller tweaks (a certain amount of copy-editing, for instance). Secondly, it has been observed that raw sequencing reads do not appear to be available through your Dryad deposition - please ensure that you check this and either make clearer how they are accessible in Dryad, or make them available through an additional alternative repository. Good luck!

RESPONSE: We thank the editor and both reviewers for looking over manuscript again and providing a second set of thorough and constructive reviews. We are glad that both reviewers are happy with our revised manuscript, and we have carefully considered the additionally requested changes. In response to the remaining concerns from both reviewers regarding linkage disequilibrium analysis, we have made substantial changes to our results and discussion and acknowledged the limitations of this approach. We have also carried out additional analyses testing how all morphological variables relate to *ADAM12* SNP genotypes. Finally, we have now provided the depository location for the .qseq file of raw RAD reads on Dryad ([https:// doi.org/10.5061/dryad.9642b](https://doi.org/10.5061/dryad.9642b)). We apologise that this link was not provided previously. We provide a point-by-point response to each reviewer comment.

Reviewer 1

COMMENT: I thank the authors for taking into account my suggestions as well as the ones from reviewer 2. I think that the manuscript has been subsequently improved and I am very happy with the answers you provided. However, in my opinion, one concern remains regarding the Linkage Disequilibrium analyses. I find them a lot clearer now that you have provided context and I have carefully read the references that you added which I think are extremely interesting. I agree with you that the Selvagens have a higher baseline LD, as the Madeiran island populations, that could suggest a reduced genetic diversity and small population size. My question concerns what you mean by “shallow” and “sharp”. When looking at Figure 2B, I see that the curves of the Madeiran islands (green) and the Selvagens (orange) are sharper than the curves of the Canary Islands (purple), since the initial slope is more negative in the curves of the Madeiran and Selvagen islands than in the curves of the Canary Islands. Yet, you say that “LD decay was shallow across the three Madeiran islands”, so I did not understand well what you mean by sharp and shallow. In addition, still concerning the LD analyses, in all the papers that you cited, simulations were conducted to confirm that the shape of the LD curve was consistent with a bottleneck. I think this would be a very interested venue, but as you already did many analyses and the paper is close to publication, maybe you could only mention that a simulation-based approach would help confirming that these LD plots can be inferred in terms of bottlenecks. Finally, another interesting analysis would be to use your r^2 estimations to infer population size and variation in population size over time from Sved Equation (1971) as in Liu et al. 2017 or Pilot et al. 2014 if you have enough data for this. I'm particularly interested in knowing if you can decipher a long term low population size because the islands are small from a recent bottleneck due to founder event as in Pilot et al. 2014.

Genomic variation, population history and within-archipelago adaptation between island bird populations

RESPONSE: We are glad that the reviewer feels we have well addressed their concerns. In terms of the LD plots, we agree that the use of ‘shallow and sharp’ was confusing, and have simplified the conclusions we make based on these results. We now only broadly describe differences in LD, and the associated in population sizes inferred from these trends. Our LD results, now read as follows, on lines 419- 423:

“LD was highest in the smallest and most isolated populations across the Berthelot’s pipit range. Thus, across all populations the Selvagens had the highest LD with a long range decay pattern; LD was lower across the three Madeiran islands and was lowest in the Canary Islands, especially in large central islands (Fig. 2B).”

Our discussion of these results now reads as follow, lines 612 - 616:

“Past colonisation history and associated bottlenecks are reflected in patterns of population level LD; rapid LD decay at proximate SNPs and low long range LD indicates larger and more outbred populations across the Canary Islands, while high long range LD indicate bottlenecks and/ or inbreeding and reduced genetic diversity (Pilot *et al.*, 2014; Shannon *et al.*, 2015).”

We agree that it would be interesting to use a simulation-based approach to further confirm these patterns or to add detail to the population size estimates at different historical time points. However, as you say we feel additional analyses will overcomplicate and clutter this manuscript where these types of analyses are likely to be better conducted using dense SNP data sets, such as those from whole genome sequences. We have added a comment on this idea in the discussion, on lines 619 - 621:

“Simulation-based approaches, using a greater density of SNPs, may be useful to further confirm if our LD patterns are as a result of population bottlenecks or to add detail to the population size estimates at different historical time points.”

COMMENT: Here are some additional minor changes that I’m suggesting. Please note that the line numbers correspond to the manuscript with track changes:

COMMENT: L 51. change is dependent to depends upon

RESPONSE: Done.

COMMENT: You removed ‘demography’ in the conclusion, so should you remove it also here?: L 271. 289.

RESPONSE: We have removed the first reference, and changed the second to “To further understand patterns of genetic diversity and population size in the Berthelot’s pipit, we estimated LD for each island population using Plink.” (now line 286 - 288).

COMMENT: L 374. the Tawny pipit rooted tree obtained without adding migration events

RESPONSE: Done.

COMMENT: L 378. including just→limited to

RESPONSE: Done.

COMMENT: Figure 1: maybe add the three colours of the different archipelagos, for example as a frame in each of the archipelago name? And in that case don’t use the green in the zooming frame

Genomic variation, population history and within-archipelago adaptation between island bird populations

as it is already used for the Madeiran archipelago.

RESPONSE: We think that adding additional colour to the map in Figure 1 could then get confusing when including the colours in Figure 3 PCA plots. We have made sure that the colour scheme is consistent between the *TreeMix* labels (Fig. 2A) and the LD plot colours (Fig. 2B) so that these plots are easily comparable.

COMMENT: Figure 2: add the Selvagens in the figure caption.

RESPONSE: Done.

COMMENT: Figures S9 and S10: indicate the chromosome Z

RESPONSE: Done.

COMMENT: 742. whose

RESPONSE: Changed.

COMMENT: I want to congratulate again the authors for this manuscript that I really much enjoyed reading and I hope that they will find my questions about linkage disequilibrium analyses relevant.

RESPONSE: Thank you for the positive comments. It is really nice to hear that you have enjoyed reading the manuscript.

Reviewer 2

COMMENT: I acted as reviewer # 2 for the initial submission of your manuscript “Genomic variation, population history and within-archipelago adaptation between island bird populations”. I think you now improved the manuscript considerably, taking also my comments and suggestions into account. I have only one major comment remaining and some minor edits:

RESPONSE: We would like to thank you again for the time taken to constructively aid the interpretation and conclusions that we have made in this manuscript, and for you extremely thorough review and code.

COMMENT: 1. LD-based inferences on N_e . I do not think that `loess()` is the most appropriate way of fitting a LD-decay function to your data. In the end, you do not get any parameters or statistics on which you could base your conclusions. Also, (1) how did you define the span-parameter (and based on what?) and (2) `loess()` is known to overreact which may explain the dip around 25–50 kb distance. Thus, I would suggest using an appropriate model here, as for example done in Knief et al. (2017): You could fit the function described in Hill & Weir (1988), which is

$$LD.data \sim \left(\frac{(10 + \rho * distance)}{(2 + \rho * distance) * (11 + \rho * distance)} \right) * \left(1 + \frac{(3 + \rho * distance) * (12 + 12 * \rho * distance + (\rho * distance)^2)}{(n * (2 + \rho * distance) * (11 + \rho * distance))} \right)$$

where n is the sample size and ρ represents the population recombination parameter ($\rho = 4 * N_e * r$) that is going to be estimated.

Given your LD data in a data frame called “out” with columns “Distance” and “Pearson_r2”, you can

Genomic variation, population history and within-archipelago adaptation between island bird populations

use the following R code:

```
out1 <- out[order(out$Distance),]
distance <- out1$Distance
LD.data<- out1$Pearson_r2
n <- 2*939
HW.st<-c(C=0.1)
HW.nonlinear <-
nls(LD.data~((10+C*distance)/((2+C*distance)*(11+C*distance)))*(1+((3+C*distance)*(12+12*C*distance+(C*distance)^2))/(n*(2+C*distance)*(11+C*distance))),start=HW.st,control=nls.control(maxiter=100))
tt <- summary(HW.nonlinear)
new.rho <- tt$parameters[1]
fpoints <-
((10+new.rho*distance)/((2+new.rho*distance)*(11+new.rho*distance)))*(1+((3+new.rho*distance)*(12+12*new.rho*distance+(new.rho*distance)^2))/(n*(2+new.rho*distance)*(11+new.rho*distance)))
```

and plot the estimated curve with
`points(distance,fpoints,col="orangered",type="l")`

You could then compare your estimates of the parameter “C”, e.g. by getting confidence intervals with
`confint(HW.nonlinear)`

I guess (but have never done so) that you could also get bootstrapped confidence bands for the fitted lines (see, for example, <https://cran.r-project.org/web/packages/nlraa/vignettes/Bootstrapping.html>).

You could also estimate the LD-half-decay, e.g. through:

```
# Choose between actual and estimated values:
```

```
HalfDecay <- max(LD.data)/2
```

```
HalfDecay <- max(fpoints)/2
```

```
down <- out1[max(which(fpoints>=HalfDecay)), ]
```

```
up <- out1[min(which(fpointshalf.decay.distance <- out1[min(which(abs(fpoints-HalfDecay)), ]
```

I think that this formula (and others) are also implemented in ngsLD (Fox et al. 2019).

I hope you find these comments helpful and best wishes.

RESPONSE: Thank you for providing the code for us to try an alternative line fitting method for the LD plots by population. When using the fitted line based on the recombination factor calculation provided, we generate some unusual results for our populations. These plots are easier to visualise when distance between SNPs are log-transformed, as presented on the graphs below. Here, we see that LD decays over long distances in the Selvagens, somewhat shorter distances in Madeira and then substantially shorter in the Canary Islands (see Tenerife example plot below), and that the half decay point reflects this (LD half decay $R^2 \sim 0.7$ SG, 0.6 M and 0.3 CI). These differences between archipelagos are large and support our results presented using the LOESS method of data fitting. Further, we would re-emphasise that these observed differences in genetic diversity across

Genomic variation, population history and within-archipelago adaptation between island bird populations

archipelagos are also consistent with previous analyses based on approximate Bayesian computation modelling, and with other genetic diversity measures (Spurgin *et al.* 2014).

LOESS is a commonly used approach to fit LD data (for examples see studies of House Finches, Shultz *et al.* 2016; cultivated tomato, Robbins *et al.* 2011; sheep breeds, Prieur *et al.* 2017 and Alpine trees, Choudhury *et al.* 2019). Our previous work (Armstrong *et al.* 2018) and many other studies use this method to fit LD decay curves and so to allow comparability to this work we retain this method in the manuscript. However, in response to this comment and the comment from Reviewer 1, we have simplified the conclusions we draw from these plots. We have also added details on the span parameter used, and have added and acknowledgment of the method fitting limitations into the discussion. We specifically acknowledge and refer to the approach outlined by the reviewer, on lines 623 - 626:

“This pattern has been found in previous studies of LD in this system using the LOESS line fitting method (Armstrong *et al.*, 2018). We are unable to determine a biological explanation for this, but alternative line fitting methods such as those used by Hill & Weir (1988) reflect our archipelago level conclusions.”

Genomic variation, population history and within-archipelago adaptation between island bird populations

Minor comments (line numbers refer to the document with track changes):

COMMENT: L84: “selection can also be”

RESPONSE: Changed.

COMMENT: L87: “may reveal patterns of strong”

RESPONSE: Changed.

COMMENT: L95: “adaptations”

RESPONSE: Changed.

COMMENT: L108: “including diseases and climatic factors”

RESPONSE: Changed.

COMMENT: L132: Do you mean divergent or disruptive selection instead of “differential”?

RESPONSE: Changed.

COMMENT: L141: “across the species range”

RESPONSE: Changed.

COMMENT: L151: “variation to variation in phenotypic”

RESPONSE: Changed.

COMMENT: L173: “and birds were molecularly”

RESPONSE: Changed.

COMMENT: L223: “population structure”

RESPONSE: Changed.

COMMENT: L228: Here and throughout: Check the tense you are writing in. I think it must be simple past here, i.e. “We implemented”

RESPONSE: Changed in the relevant places in the methods section.

COMMENT: L253: “analyses”

RESPONSE: Changed.

COMMENT: L265: Here and throughout: Do you mean “tree topology” instead of “tree topography”?

RESPONSE: Changed throughout.

COMMENT: L265: “pipit populations, we computed”

RESPONSE: Done.

COMMENT: L281: “larger, more outbred populations show a rapid”

RESPONSE: Changed.

COMMENT: L288: “Ne, and LD at”

RESPONSE: Changed.

COMMENT: L290: “calculated” or “estimated”

RESPONSE: Changed.

Genomic variation, population history and within-archipelago adaptation between island bird populations

COMMENT: L294: “the R function”. How did you set the span parameter?

RESPONSE: Done. We have added:

“the R function ‘loess’ using the default span parameter (0.75)”

COMMENT: L305: “LD, which enabled us”

RESPONSE: Changed.

COMMENT: L306: “implemented in”

RESPONSE: Changed.

COMMENT: L309: “individually for an association with these axes”

RESPONSE: Done.

COMMENT: L313: I guess you either mean above the $-\log_{10}(P)$ threshold or below the P -value threshold.

RESPONSE: This refers to the Bonferroni corrected significance P threshold described in the previous sentence.

COMMENT: L318-320: Delete that sentence, it is repetitive with the above.

RESPONSE: Done.

COMMENT: L325: “After having”

RESPONSE: Done.

COMMENT: L329: Please add version numbers

RESPONSE: Done.

COMMENT: L329: “implemented in R”

RESPONSE: Changed.

COMMENT: L332: “SNPs, we fitted LMMs [...] as dependent variables.”

RESPONSE: Done.

COMMENT: L332: Why do you not test all other phenotypic traits, e.g. wing length?

RESPONSE: We have now provided summarised results for all morphological variables measured as requested on lines 557- 571:

“To determine what effect candidate SNP variation may have on morphology, we tested for genotype associations with wing, tarsus and head length, weight and bill length, width and height..... Genotype was not significantly associated with beak morphology variables (bill length, height or width), weight, wing length or tarsus length, although there were differences between sexes for tarsus length ($P < 0.003$), wing length ($P < 0.002$) and bill length ($P < 0.010$) as we expect for a sexually dimorphic species.”

COMMENT: L347: “tree, we trimmed”

RESPONSE: Done.

COMMENT: L366: “showed”

RESPONSE: Done.

Genomic variation, population history and within-archipelago adaptation between island bird populations

COMMENT: L375: “explained” and “variance explained”

RESPONSE: Done.

COMMENT: L378: “placed”

RESPONSE: Done.

COMMENT: L388: “especially in the central”

RESPONSE: Done.

COMMENT: L389: “highest” instead of “strongest”

RESPONSE: Done.

COMMENT: L401: “suggested few”

RESPONSE: Done.

COMMENT: L402: “occurred” instead of “exist”?

RESPONSE: Done.

COMMENT: L407: “smallest and most isolated”

RESPONSE: Done.

COMMENT: L411: You could add the half-decay distances here.

RESPONSE: See comment above. We have removed specific reference to the shape of the curves.

COMMENT: L455: “6,252 bases downstream”

RESPONSE: Done.

COMMENT: L492: I do not think that “Analysis of the first” is correct here. It is just the plot of the 1st PC vs 2nd PC.

RESPONSE: Done.

COMMENT: L537: “genotype associations”

RESPONSE: Done.

COMMENT: L538: If you do not use all phenotypes then delete them from the methods section. However, I strongly encourage you to present all results, including those for wing length and weight.

RESPONSE: We have now presented GLMM results for all seven of the morphological variables, as detailed above, and so have kept this section in the methods as you suggest.

COMMENT: L546: “putatively under selection”

RESPONSE: Changed.

COMMENT: L577: I think it was “head length”, not “beak length”.?

RESPONSE: Changed.

COMMENT: L578: “and for a SNP in a candidate gene”

RESPONSE: Done.

COMMENT: L587: This needs a quantitative assessment, as described in my major point above.

Genomic variation, population history and within-archipelago adaptation between island bird populations

RESPONSE: We have removed the emphasis on “shallowness and sharpness” of the LD curves (see response to Reviewer 1), but instead highlight the key overall finding that LD is highest in the Selvagens, followed by the Madeiran islands and then the Canary Islands. These broad conclusions are consistent irrespective of the line fitting method used but we agree that we are unable to provide clarity with the results presented here on population level bottlenecks.

COMMENT: L598: “larger and more outbred”

RESPONSE: Done.

COMMENT: L605: What are these previous studies?

RESPONSE: Reference to Armstrong *et al.* (2018) added, where LD was investigated across pipit archipelagos.

COMMENT: L640: “putatively under divergent”

RESPONSE: Changed.

COMMENT: L654: “may be under selection”

RESPONSE: Changed.

COMMENT: L662-663: I think this is not correct. Or did you control for size somehow?

RESPONSE: Sorry this has now been changed to match the simplified GLMM results, referring to head length.

COMMENT: L668: “history, there”

RESPONSE: Done.

COMMENT: L682: “under selection in recently diverged”

RESPONSE: Done.

COMMENT: L734: Please provide also the doi/accession number for the raw sequencing data

RESPONSE: Done. We have added details on how to access the raw ddRAD reads file:

“The .qseq file of raw RAD reads for each individual sample are also available on Dryad ([https:// doi.org/10.5061/dryad.9642b](https://doi.org/10.5061/dryad.9642b)).”

References

Armstrong, C., Richardson, D.S., Hipperson, H., Horsburgh, G.J., Küpper, C., Percival-Alwyn, L., Clark, M., *et al.* (2018). Genomic associations with bill length and disease reveal drift and selection across island bird populations. *Evolution Letters* **2**:22–36.

Choudhury, R.R., Rogivue, A., Gugerli, F. and Parisod, C. (2019). Impact of polymorphic transposable elements on linkage disequilibrium along chromosomes. *Molecular Ecology* **28**:1550–1562.

Prieur, V., Clarke, S.M., Brito, L.F., McEwan, J.C., Lee, M.A., Brauning, R., Dodds, K.G., *et al.* (2017). Estimation of linkage disequilibrium and effective population size in New Zealand sheep using three different methods to create genetic maps. *BMC Genetics* **18**:1–19.

Robbins, M.D., Sim, S.C., Yang, W., Van Deynze, A., Van Der Knaap, E., Joobeur, T. and Francis, D.M.

Genomic variation, population history and within-archipelago adaptation between island bird populations

(2011). Mapping and linkage disequilibrium analysis with a genome-wide collection of SNPs that detect polymorphism in cultivated tomato. *Journal of Experimental Botany* **62**:1831–1845.

Shultz, A.J., Baker, A.J., Hill, G.E., Nolan, P.M. and Edwards, S. V. (2016). SNPs across time and space: population genomic signatures of founder events and epizootics in the House Finch (*Haemorrhous mexicanus*). *Ecology and Evolution* **6**:7475–7489.